# IMPA1 dependent regulation of phosphatidylinositol 4,5-bisphosphate and calcium signalling by lithium

Sankhanil Saha ⓘ, Harini Krishnan, Padinjat Raghu ⓘ

**Lithium (Li) is widely used as a mood stabilizer to treat bipolar affective disorder. However, the molecular targets of Li that underpin its therapeutic effect remain unresolved. Inositol monophosphatase (IMPA1) is an enzyme involved in phosphatidylinositol 4,5-bisphosphate (PIP$_2$) resynthesis after PLC signaling. In vitro, Li inhibits IMPA1, but the relevance of this inhibition within neural cells remains unknown. Here, we report that treatment with therapeutic concentrations of Li reduces receptor-activated calcium release from intracellular stores and delays PIP$_2$ resynthesis. These effects of Li are abrogated in *IMPA1* deleted cells. We also observed that in human forebrain cortical neurons, treatment with Li reduced neuronal excitability and calcium signals. After Li treatment of human cortical neurons, transcriptome analyses revealed down-regulation of signaling by glutamate, a key excitatory neurotransmitter in the human brain. Collectively, our findings suggest that inhibition of IMPA1 by Li reduces receptor-activated PLC signaling and neuronal excitability.**

## Introduction

Bipolar affective disorder (BPAD) is a human psychiatric illness characterized by disruptive mood swings; the defining characteristics being the alteration between mania and depression associated with an elevated rate of suicide among patients (Osby et al, 2001). BPAD has been regarded by the World Health Organization as one of the leading causes of disability worldwide (Lopez & Murray, 1998). Mood stabilizers like lithium (Li), carbamazepine, and valproate remain the first-line mode of medication for bipolar disorder. Among these drugs, Li stands out as the go-to drug for treating acute manic phases and for its preventive effect against relapses of manic episodes (Geddes et al, 2004; Cipriani et al, 2013). Although Li remains the mainstay in BPAD treatment, individual patient responses towards Li are variable—it remains ineffective for a large proportion (~30%) of BPAD patients, the so-called "Li non-responders," whereas another 30% of patients are only partially responsive to Li (Hou et al, 2016).

The reason for this variable response is unknown and remains a topic of intense interest. Despite substantial research over several decades, the molecular and biochemical mechanism by which Li exerts its effects on human brain cells that are relevant to functional improvements remains unclear (Haggarty et al, 2021). Many molecular targets for Li have been described (Roux & Dosseto, 2017). These include inositol monophosphatase (IMPase) (Harwood, 2005), glycogen synthasekinase-3$\beta$ (GSK-3$\beta$) (Freland & Beaulieu, 2012), sodium myo-inositol co-transporter (Dai et al, 2016), and bisphosphate-3'-nucleotidase (BPNT1) (Spiegelberg et al, 2005). The two most well-studied targets of Li are IMPase and GSK-3$\beta$.

During GPCR signalling, many receptors activate PLC leading to the hydrolysis of the signalling lipid phosphatidylinositol 4,5-bisphosphate (PIP$_2$) generating inositol 1,4,5-trisphosphate (IP$_3$) and DAG (Berridge, 2009), ultimately leading to changes in intracellular calcium [Ca$^{2+}$]$_i$. This mechanism of cell signalling is used by numerous G-protein-coupled receptors in the brain that participate in synaptic transmission such as glutamate and acetyl choline. Previous studies in rat hippocampal cultures has shown that treatment with Li diminishes Ca signalling responses to Gq-PLC–coupled neurotransmitter receptors such as the metabotropic glutamate receptor and muscarinic acetylcholine receptor (Sourial-Bassillious et al, 2009) and alterations in [Ca$^{2+}$]$_i$ have been reported in cell lines derived from BPAD patients (Wasserman et al, 2004). Most recently, a meta-analysis has revealed that basal levels of [Ca$^{2+}$]$_i$ and [Ca$^{2+}$]$_i$ responses to agonist stimulation in BPAD patients are elevated (Harrison et al, 2021). Thus, a link between Ca signalling and BPAD has been previously noted although the underlying mechanism in this context remains unclear.

A long-standing hypothesis for the therapeutic effect of Li is its impact on inositol lipid turnover during receptor-activated G-protein-coupled signalling through inhibition of IMPase (Harwood, 2005). PIP$_2$ is a low-abundance membrane lipid and its resynthesis is required to sustain Ca signalling during high rates of PLC activity as is seen in the brain (Yang et al, 2016) during the activation of many GPCR required for neurotransmission such as the metabotropic glutamate receptor (Reiner & Levitz, 2018). The resynthesis of PIP$_2$ occurs via a set of biochemical reactions referred to as the PIP$_2$ cycle (Cockcroft & Raghu, 2016). The sequential dephosphorylation of IP$_3$ produced during PLC activity to generate inositol is required for PIP$_2$ resynthesis

National Centre for Biological Sciences-TIFR GKVK Campus, Bangalore, India

Correspondence: praghu@ncbs.res.in

(Fig 1A). IMPase is the enzyme required for the final step in the conversion of inositol 1-phosphate to inositol. Biochemical analyses showed a decrease in myo-inositol concentration in cerebral cortex of lithium-treated rats accompanied by an increase in the concentration of inositol monophosphate suggesting that Li might inhibit IMPase (Allison & Stewart, 1971; Allison et al, 1976). Subsequent studies showed that Li binds to IMPase and inhibits the enzyme in an uncompetitive manner (Allison et al, 1976; Hallcher & Sherman, 1980; Gee et al, 1988). These observations led Berridge to propose the "Inositol depletion hypothesis" (Berridge et al, 1989) which posits that during PLC signalling in the brain, the inhibition of IMPase by Li restricts the supply of inositol for PIP$_2$ resynthesis leading to down regulation of neurotransmitter receptor signalling, thus reducing excitability in the brain and the control of mania. However, the link between G-protein-coupled PIP$_2$ turnover, $[Ca^{2+}]_i$ signalling and the action of Li in human neurons remains to be established.

The mammalian genome includes two genes encoding IMPase activity, *IMPA1*, and *IMPA2*. However, IMPA1 is more abundant in the human brain (BrainSeq A Human Brain Genomics Consortium et al, 2015) and is inhibited by Li at therapeutically relevant concentrations (Ohnishi et al, 2007). In vivo studies have also suggested a role of Li in regulating IMPase function. Briefly, (i) in rodent models, Li treatment reduced brain inositol levels and elevated inositol monophosphate (IP$_1$), the substrate of IMPase in the cerebral cortex of rats (Sherman et al, 1981); (ii) treatment of human BPAD patients with Li led to a reduction in PIP$_2$ levels in platelet membranes (Soares et al, 1997, 1999, 2000). However, there are also findings that do not support this hypotheses (i) *IMPA1* knockout mice showed only a modest reduction in brain inositol levels (Cryns et al, 2008) and the impact of *IMPA1* knockout on receptor activated PIP$_2$ turnover remains unknown, (ii) genome-wide association studies have not revealed polymorphisms in IMPA1 that are linked to BPAD or Li sensitivity (Sjoholt et al, 2000; Hou et al, 2016), (iii) Li-mediated inhibition of IMPase leads to an accumulation of IP$_1$ and this elevated IP$_1$ can lead to an inhibition of the rate of PI synthesis and hence PIP$_2$ (Saiardi & Mudge, 2018).

In this study, we have tested the hypothesis that treatment of human cells with Li diminishes calcium signalling. We find that this change is underpinned by reductions in G-protein-coupled PIP$_2$ turnover. These changes in calcium signalling were also observed in human-induced pluripotent stem cell (iPSC)-derived forebrain cortical neurons and accompanied by reduction in neuronal excitability. Transcriptomic analysis revealed that Li treatment exerts its effect on neuronal excitability through multiple mechanisms including but not restricted to GPCR-activated calcium signalling. Our findings open new approaches to predict Li responsiveness in BPAD patients and develop novel strategies for the clinical management of this disorder.

## Results

### An in vivo model system to study the impact of Li on PLC-induced PIP$_2$ turnover

To study the turnover of PIP$_2$ after receptor-activated PLC turnover, we created a cell line which heterologously expressed two components. To stimulate high levels of PLC activity, we expressed

the human muscarinic acetylcholine receptor (m1AchR) encoded by the gene *CHRM1*; previous studies have shown that activation of heterologously expressed m1AchR results in high rates of PIP$_2$ hydrolysis leading to IP$_3$ production (Parker et al, 1991; Raghu et al, 1997). To monitor the levels of PIP$_2$ at the plasma membrane, we expressed the PH domain of PLC-$\delta$ fused to GFP (PH-PLC$\delta$::GFP). This probe binds to PIP$_2$ and in imaging experiments; its relative distribution between the plasma membrane and the cytosol has been used by us and others to monitor plasma membrane PIP$_2$ levels (Várnai & Balla, 1998; Chakrabarti et al, 2015). The PH-PLC$\delta$-GFP probe is a sensitive reporter for PIP$_2$ but it also binds to IP$_3$; there is another probe – Tubby c-terminal domain that is also employed as a sensor for PIP$_2$. However, the Tubby c-terminal domain has exhibited lower sensitivity to report on changes of PIP$_2$ during PLC activation, although being more specific in its affinity towards PIP$_2$ (Szentpetery et al, 2009). Furthermore, a very recent study has noted that in contrast to PH-PLC$\delta$-GFP probe, the Tubby domain binds selectively to certain domains of the plasma membrane at membrane contact sites making it not an optimal detector of PIP$_2$ levels across the plasma membrane (Thallmair et al, 2023). Due to these reasons, we preferred to use the PH-PLC$\delta$-GFP probe as the sensor for PIP$_2$.

A cell line expressing both m1AchR and PH-PLC$\delta$::GFP can therefore be used to stimulate PLC and monitor the turnover of PIP$_2$ at the plasma membrane. We generated stable HEK293T cell lines expressing m1AchR (control-M1) and PH-PLC$\delta$::GFP was transiently expressed in those cells (Fig 1B); when stimulated with the m1AchR agonist carbamoylcholine (carbachol: Cch), the levels of PH-PLC$\delta$:: GFP at the plasma membrane drop after which they gradually recover back towards pre-stimulation levels (Fig 1C). The changes in plasma membrane PIP$_2$ was quantified as the ratio of fluorescence intensity of the PH-PLC$\delta$::GFP probe at the plasma membrane to its fluorescence intensity in the underlying cytosol in the same part of the cell. As a positive control for our assay to monitor PIP$_2$ regeneration after PLC activation, we used shRNA to deplete Nir2; Nir2 (PITPNM1/RdgB$\alpha$I) encodes for a phosphatidylinositol transfer protein required for the resynthesis of PIP$_2$ after PLC stimulation (Kim et al, 2013, 2016; Chang & Liou, 2015). Under our experimental conditions, we found that depletion of Nir2 in our HEK293T reporter line (Fig S1A) resulted in a decreased rate in the recovery of plasma membrane PIP$_2$ levels after stimulation with carbachol (Figs 1D and E and S1B).

To measure the effect of Li treatment on PIP$_2$ resynthesis, we incubated the reporter line with 1 mM Li; this concentration of LiCl was used in our experiments because the therapeutic range for Li medication in BD patients was 0.75–1.5 mEq/litre (Francis et al, 2004). After Li treatment, we stimulated the cells with carbachol and compared the dynamics of plasma membrane PIP$_2$ levels with that in non-Li–treated cells. We found that in Li-treated cells, the rate with which PIP$_2$ levels at the plasma membrane recovered to pre-stimulation levels was significantly slower than in untreated controls (Fig 1F and G). These findings strongly suggest that treatment with Li results in a reduction in the rate of PIP$_2$ resynthesis after PLC stimulation.

In an alternative approach, we quantified total cellular PIP$_2$ mass using liquid chromatography coupled with tandem mass spectrometry (LC-MS/MS) at three distinct time points—corresponding to the basal state, one-minute post-stimulation of PLC via

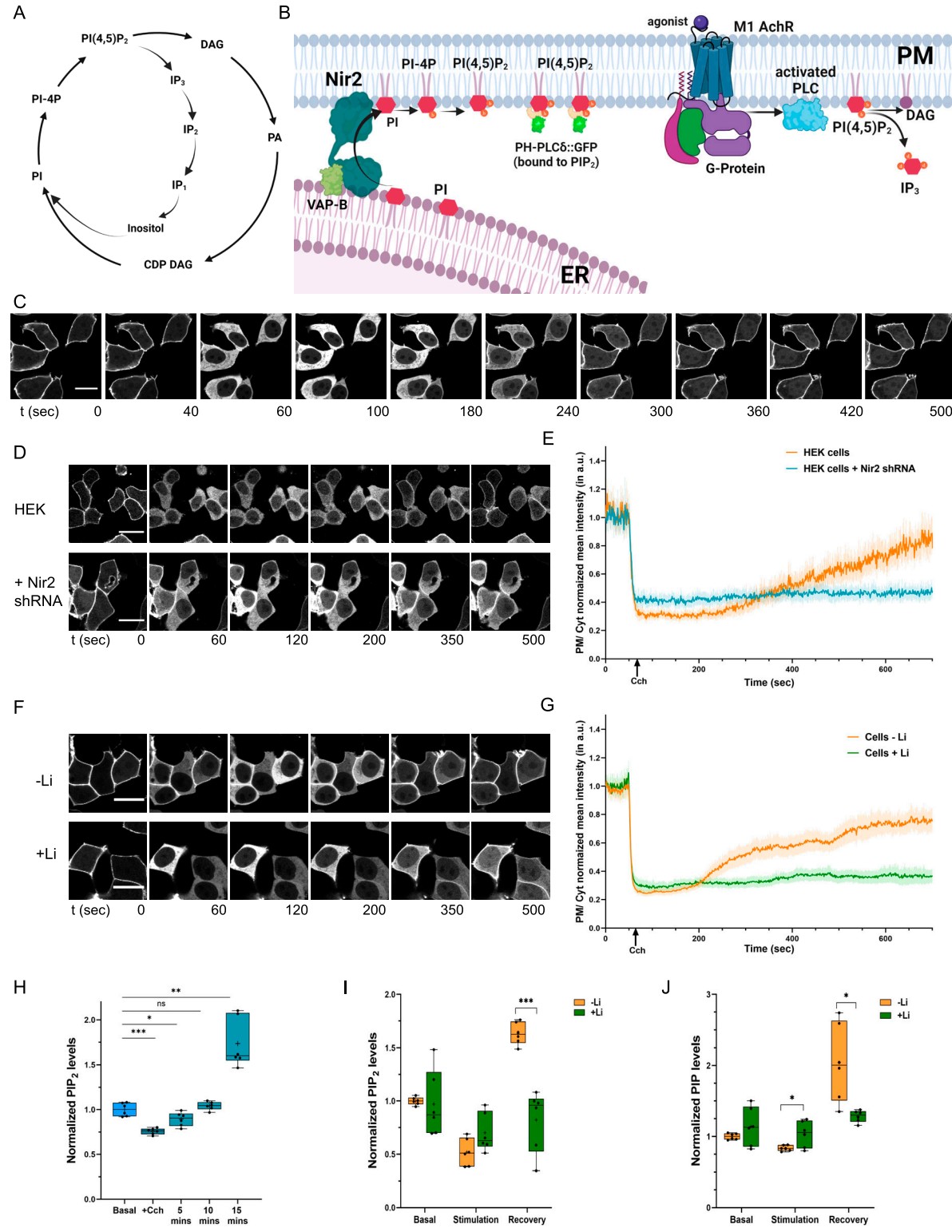

**Figure 1. An in vivo model system to study the impact of Li on PLC-induced phosphatidylinositol 4,5-bisphosphate (PIP₂) turnover.**
**(A)** The phosphatidylinositol bisphosphate (PIP₂) cycle. **(B)** Schematic representation of an in vivo system elucidating the role of Nir2 and m1AchR in the PIP₂ cycle and the PH-PLCδ::GFP probe binding to the PIP₂ at the plasma membrane. **(C)** The PH-PLCδ::GFP probe translocates to the cytoplasm and back to the plasma membrane, corresponding to the hydrolysis and regeneration of PIP₂, respectively (control–M1; HEK293T cells expressing m1AchR). **(D, E)** Nir2 depletion leads to a delayed regeneration of PIP₂ at the plasma membrane. Changes in membrane localization of the PH-PLCδ::GFP probe after Cch stimulation (denoted by the ↑) are quantified via the fluorescence intensity ratio of the plasma membrane and the cytosol (PM/Cyt) at various times. This ratio is normalized to the first time-point and the mean ± 95% C.I. is plotted (from three experiments each performed in replicates) for control cells (orange line, n = 33), and for cells treated with Nir2 shRNA (blue line, n = 28). **(F, G)** Rate of

carbachol addition (hydrolysis of $PIP_2$) state, and post-stimulation (recovery of $PIP_2$) state. From these experiments, the time for recovery of $PIP_2$ (to pre-stimulation levels) was standardized at 15 min post stimulation with carbachol (Fig 1H). Using these conditions, we compared the total $PIP_2$ mass between untreated cells and those treated with Li. We found that the mass of $PIP_2$ during the recovery phase was lower in Li-treated cells compared with controls (Figs 1I and S1C); likewise, the level of PIP in Li-treated cells was also lower during the recovery phase (Figs 1J and S1D). Thus, Li treatment lowers the recovery of PIP and $PIP_2$ levels after PLC activation. PIP refers collectively to all the isoforms of PIP: PI3P, PI4P, and PI5P. These cannot be individually distinguished by mass alone because their mass is identical. However, preexisting literature has shown that PI4P is the most abundant isoform of PIP present in cells; its level is ~50-fold higher than that of PI5P (Rameh et al, 1997). Hence, we can infer that the change in the total mass of PIP observed by LC-MS/MS is mainly reflective of the levels of PI4P.

### Chronic Li treatment reduces PLC-dependent intracellular $Ca^{2+}$ mobilization

A key outcome of agonist-triggered, PLC-mediated $PIP_2$ hydrolysis is intracellular $Ca^{2+}$ signalling (Berridge, 2009). Because we noted a reduction in the turnover of $PIP_2$ after Li treatment, we tested the impact of Li treatment on intracellular $Ca^{2+}$ levels $[Ca^{2+}]_i$. Using the control-M1 cell line, we monitored $[Ca^{2+}]_i$ under both resting conditions and stimulation with carbachol, an agonist of m1AchR. Cells were incubated in 1 mM Li before the experiment: basal $[Ca^{2+}]_i$ were no different between treated and untreated cells (Fig 2A). Stimulation of untreated cells with carbachol evoked a rise in $[Ca^{2+}]_i$ which then decayed back to resting levels (Fig 2B). In cells treated with Li, the carbachol-induced rise in $[Ca^{2+}]_i$ was reduced compared with that seen in untreated cells (Fig 2C). Receptor-activated PLC-mediated $Ca^{2+}$ signalling is a biphasic process with two components, an initial release of $Ca^{2+}$ from intracellular stores followed by store-operated influx of $Ca^{2+}$ into the cell from the extracellular medium (Prakriya & Lewis, 2015). We performed experiments to separately monitor each of these components. During the initial phase of the assay, carbachol stimulation was performed in the presence of zero extracellular $Ca^{2+}$, thus selectively monitoring the release of $Ca^{2+}$ from intracellular stores (Fig 2D and E). After this, the extracellular solution was supplemented with $Ca^{2+}$ and the influx of $Ca^{2+}$ into the cytoplasm was monitored (Fig 2D and F). This analysis revealed that in Li-treated cells, both intracellular $Ca^{2+}$ release (Fig 2E) and the subsequent store operated $Ca^{2+}$ influx into cells (Fig 2F) was reduced compared with Li-untreated controls. This reduction in receptor-activated release of $Ca^{2+}$ from intracellular stores was not because of a reduction in the size of intracellular $Ca^{2+}$ stores as emptying of stores by the application of thapsigargin resulted in equivalent rises in $[Ca^{2+}]_i$ in Li-treated cells and untreated controls (Fig 2G and H).

According to the "inositol depletion hypothesis" (Berridge et al, 1989), Li can restrict the available cytoplasmic inositol pool available for phosphatidylinositol (PI) (and thence $PIP_2$) resynthesis by inhibiting IMPase—thereby, lowering down the synthesis of the pool of $PIP_2$ that is required for PLC activity. This cytoplasmic inositol pool can also be replenished by transport of inositol from the extracellular medium via plasma membrane inositol transporters (summarized in Balla [2013]). This implies that if a deficit in the available pool of cytosolic inositol is responsible for the attenuated calcium signalling in Li-treated cells, this might be rescued by supplementation of the extracellular medium with inositol. The standard concentration of inositol in DMEM High Glucose (DMEM; Life Technologies) is 40 $\mu$M (7 mg/litre). We grew HEK293T cells in an inositol-rich DMEM media; inositol concentration was ca. 28 mg/litre; which is similar to the inositol concentration in the cerebrospinal fluid (Nixon, 1955; Swahn, 1985; Shetty et al, 1996). For our inositol-rich DMEM, 117 $\mu$l of 100 $\mu$M of inositol (18 mg/ml) was added to 100 ml of DMEM High Glucose, supplemented with 10% FBS—this led to the final effective concentration of inositol in the media ~155 $\mu$M (28 mg/litre). This media were referred to as the inositol-rich media and used for the inositol supplementation experiments. These cells were subjected to Li treatment and agonist induced $Ca^{2+}$ influx was quantified. As controls in this experiment, we used Li-treated control-M1 cells grown in DMEM not supplemented with inositol. As in previous experiments, we found that when cells were grown in DMEM, Li treatment resulted in a reduction of agonist-induced $Ca^{2+}$ influx (Fig 2I and J). However, when cells grown in inositol-supplemented DMEM were treated with Li, the Li-induced reduction in $Ca^{2+}$ influx was rescued compared with that seen in cells grown under equivalent Li treatment but in DMEM not supplemented with inositol (Fig 2I and J), compared with Li non-treated cells. We also observed that the decreased rate of $PIP_2$ resynthesis post PLC stimulation in Li-treated cells was rescued by supplementation of inositol in the medium (Fig S2D).

### IMPA1 is required for agonist-mediated calcium signalling and $PIP_2$ resynthesis

IMPase, the enzyme that dephosphorylates inositol monophosphate to generate myo-inositol has been proposed as a direct target of the action of Li, inhibiting its activity (Hallcher & Sherman, 1980; Saudek et al, 1996). There are two genes encoding IMPase in the human genome *IMPA1* and *IMPA2* of which *IMPA1* is the predominant isoform expressed in HEK293T cells. To test the requirement of *IMPA1*, we generated a HEK293T cell line in which *IMPA1* was deleted by CRISPR/Cas9 genome editing (Fig S3A); m1AchR was expressed in these cells by lentiviral transduction; these cells will be denoted as IMPA1$^{-/-}$M1. Western blot analysis of IMPA1$^{-/-}$M1 showed the complete absence of the IMPA1 protein

---

regeneration of $PIP_2$ at the plasma membrane is shown for control–M1 cells (HEK293T cells expressing m1AchR) subjected to lithium treatment, compared with untreated cells. Mean ± 95% C.I. is plotted from four experiments, each performed in replicates (untreated cells—orange line, n = 58; cells treated with Li—green line, n = 53). (Scale bar: 20 $\mu$M). **(H)** Total $PIP_2$ levels recovered after various time intervals post stimulation with Cch. **(I, J)** Total $PIP_2$ and PIP levels using LC-MS from whole-cell lipid extract of untreated and lithium treated cells (n = 6 for both). **(I, J)** (Statistical test: (H) one-way ANOVA with post hoc Tukey's multiple pairwise comparisons. *P-value < 0.05; **P-value < 0.01; ***P-value < 0.001; ****P-value < 0.0001 and (I, J) Multiple unpaired t test. *P-value < 0.05; **P-value < 0.01; ***P-value < 0.001).

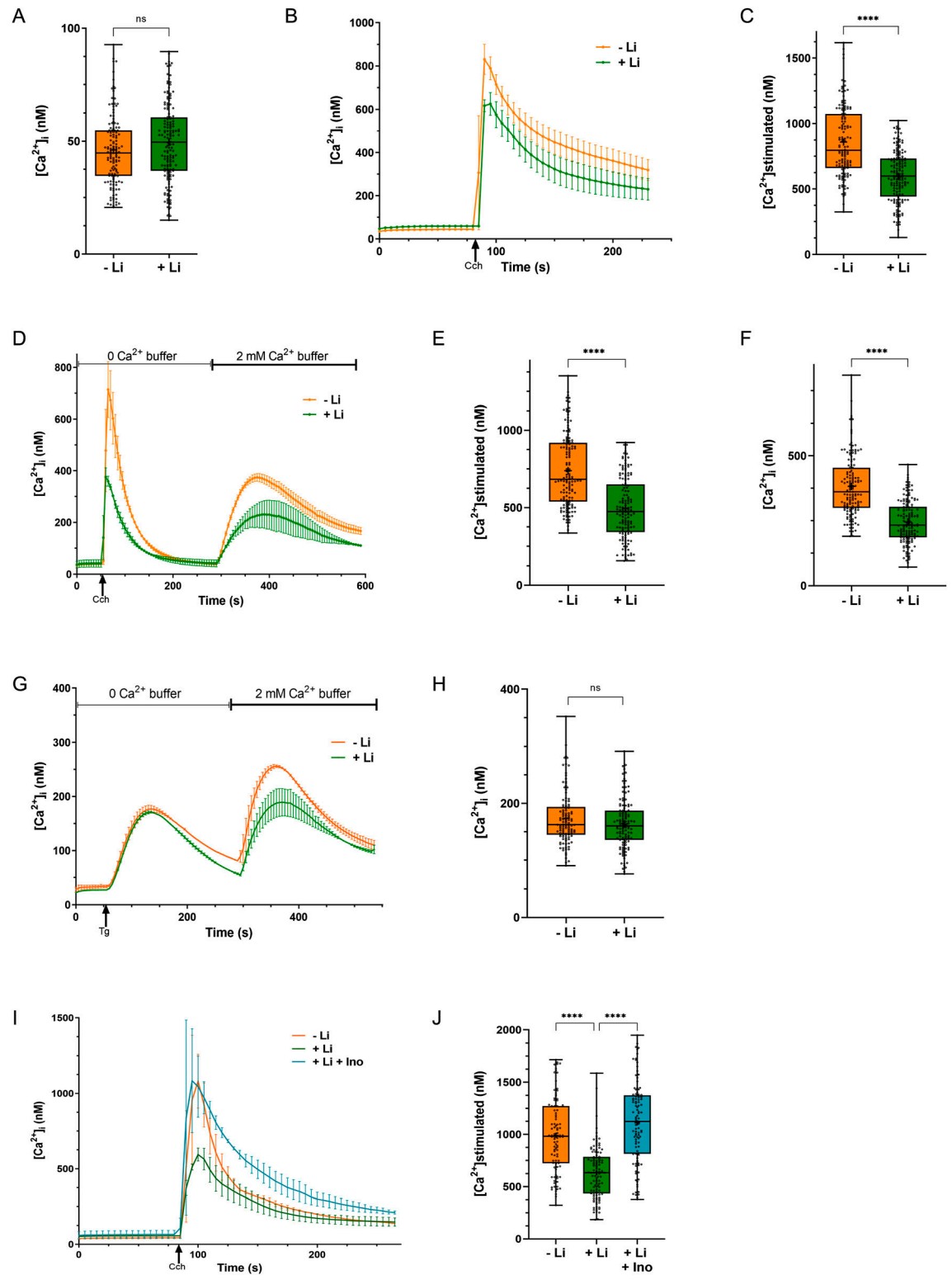

**Figure 2. Chronic lithium treatment reduces PLC dependent intracellular Ca²⁺ mobilization.**
**(A)** Basal $[Ca^{2+}]_i$, calculated in nM using Grynkiewicz equation, is unchanged in control–M1 cells because of lithium treatment (untreated cells—represented by orange, n = 133; cells treated with Li—represented by green line, n = 156). **(B, C)** PLC-dependent calcium mobilization is decreased in lithium-treated cells. For the stimulation of PLC, 20 $\mu$M of Cch was added. Quantification is done from multiple experiments, each with replicates (untreated cells [orange], n = 133; cells treated with Li [green], n = 156). **(D, E, F)** In zero Ca²⁺ buffer, agonist-dependent calcium mobilization is reduced, and subsequent Store Operated Calcium Entry is also decreased in lithium-treated cells. Quantification: untreated cells (orange), n = 145; cells treated with Li (green), n = 146. **(G, H)** Intracellular store measured by Tg, is unaltered by lithium treatment;

(Fig 3A), IMPA2 transcript levels were not altered (Fig S3B) and the levels of m1AchR protein were comparable between control–M1 cells and IMPA1$^{-/-}$M1 cells (Fig S3C and D). We compared the response of IMPA1$^{-/-}$M1 cells with control–M1 cells. When stimulated with carbachol, IMPA1$^{-/-}$M1 cells showed a reduced Ca$^{2+}$ influx response that could be rescued by reconstitution of the cells with the IMPA1 cDNA (Fig 3A–C). Further analysis under conditions of zero extra-cellular Ca$^{2+}$ revealed that the release of Ca$^{2+}$ from intracellular stores was reduced in in IMPA1$^{-/-}$M1 cells compared with control cells (Fig 3D and E) with an intact *IMPA1* gene. We also noted a decrease in the subsequent store operated calcium entry (SOCE) in these IMPA1$^{-/-}$ cells with respect to the control–M1 cells (Fig S3E). Lastly, we found that after carbachol stimulation, the rate at which PIP$_2$ levels were restored at the plasma membrane was slower in IMPA1$^{-/-}$M1 and this could be rescued by reconstitution with a WT *IMPA1* transgene (Fig 3F).

### IMPA1 is required for the effect of Li on receptor-activated Ca$^{2+}$ influx and PIP$_2$ resynthesis

We tested the effect of Li treatment on IMPA1$^{-/-}$M1 compared with untreated IMPA1$^{-/-}$M1 cells. We analysed receptor-activated Ca$^{2+}$ influx after Li treatment; this was not altered compared with un-treated IMPA1$^{-/-}$M1 cells (Fig 3G and H); in the same experiment, treatment of control–M1 cells with Li resulted in a big reduction in agonist-activated Ca$^{2+}$ influx. We also analyzed the two phases of receptor activated Ca$^{2+}$ influx and found that whereas release of [Ca$^{2+}$]$_i$ from stores was not affected by Li treatment in IMPA$^{-/-}$M1 cells, the reduction of SOCE previously noted on Li treatment in control–M1 cells was still present (Fig 3I–K). We also found that the rate of regeneration of PIP$_2$ at the plasma membrane was un-changed in IMPA1$^{-/-}$M1 cells subjected to Li treatment, compared with untreated IMPA1$^{-/-}$M1 cells (Figs 3L and S3F and G). These observations suggest that the ability of Li to inhibit Ca$^{2+}$ influx and slow PIP$_2$ resynthesis after PLC activation requires an intact IMPA1.

### Li reduces excitability in human cortical neurons

To test the relevance of our findings to human cortical neurons, changes in whose activity presumably underlies the therapeutic effect of Li in bipolar disorder patients, we generated human forebrain cortical neurons from iPSC in vitro (Sharma et al, 2020). We differentiated forebrain cortical neurons from neural stem cells (NSC) derived from an Indian control iPSC line, D149 (Fig 4A) (Iyer et al, 2018). The generated NSCs expressed previously described markers characteristic of NSC such as Nestin, SOX1, SOX2, and PAX6 (Fig 4B) and when differentiated, expressed the neuronal marker MAP2; a proportion of cells in the culture expressed the glial marker GFAP (Fig 4C). When these NSCs were differentiated into cortical neurons, they exhibited spontaneous [Ca$^{2+}$]$_i$ elevations, referred to

as transients, that increase as a function of age in vitro (Sharma et al, 2020; Akhtar et al, 2022). We measured somatic [Ca$^{2+}$]$_i$ tran-sients from such neuronal cultures at 45 d in vitro (DIV45) and quantified their frequency. We observed that the frequency of [Ca$^{2+}$]$_i$ transients was decreased after Li treatment (1 mM) for 10 d (Fig 4D and E). We also measured changes in [Ca$^{2+}$]$_i$ after stimulation with carbachol in DIV45 cortical neurons treated with Li. In our cultures, carbachol stimulation results in a rise in [Ca$^{2+}$]$_i$ that peaks before gradually decaying towards the baseline; in Li-treated cultures, the peak of this response was reduced (Fig 4F and G). We also performed experiments to measure Ca$^{2+}$ mobilization from internal stores; these revealed that carbachol-activated release of Ca$^{2+}$ in Li-treated neurons was reduced compared with those that were untreated (Fig 4H–J). By contrast, there was only a modest difference in thapsigargin-mediated rise in [Ca$^{2+}$]$_i$ be-tween Li-treated and -untreated cortical neurons (Fig S4A–C). These observations suggest that both neuronal excitability and PLC-mediated Ca$^{2+}$ signalling is attenuated in human cortical neurons after Li treatment.

### Inhibition of GSK-3$\beta$ does not phenocopy the effect of Li on human cortical neurons

It has been proposed that the inhibition of GSK-3$\beta$ by Li underlies its therapeutic effects in BPAD. To test this hypothesis, we treated HEK293T cells with the well-established pharmacological inhibitor of this enzyme- CHIR99021 (An et al, 2014). Treatment of control-M1 cells with 10 $\mu$M CHIR99021 for 24 h resulted in a robust elevation of $\beta$-catenin levels indicating potent inhibition of GSK-3$\beta$ under these conditions (Fig 5A). Under these conditions, we found that GSK-3$\beta$ inhibition did not result in reduced agonist-induced Ca$^{2+}$ influx (Fig 5B and C). Likewise, in human iPSC-derived forebrain cortical neurons, the frequency of spontaneous Ca$^{2+}$ transients (Fig 5D and E) and carbachol-activated Ca$^{2+}$ influx was not altered by GSK-3$\beta$ inhibition (Fig 5F and G). Lastly, after PLC activation and PIP$_2$ de-pletion, the rate at which PIP$_2$ levels at the plasma membrane were restored was not impacted by treatment of cells with the GSK-3$\beta$ inhibitor (Fig 5H).

### Li treatment induces transcriptional changes in pathways involved in neuronal excitability

To understand the molecular mechanisms underlying the reduced excitability after Li treatment in human iPSC-derived cortical neu-rons, we performed a transcriptomic analysis comparing untreated DIV45 neurons with Li-treated DIV45 neurons (Fig 6A). Differentially expressed genes were obtained using DeSeq2; a cut-off of log$_2$FC change greater than 0.2 was used along with a *P*-value and FDR significance at less than 0.05 were considered. Using these criteria, 335 up-regulated and 361 down-regulated genes were obtained

---

10 $\mu$M of thapsigargin was used to deplete the intracellular stores. Quantification: untreated cells (orange), n = 114; cells treated with Li (green), n = 118. **(I, J)** Inositol supplementation reversed the lithium mediated decrease in intracellular Ca$^{2+}$ release. Quantification: untreated cells (orange), n = 98; cells treated with Li (green), n = 114, cells treated with Li and high inositol (blue), n = 104. All the quantification graphs are represented by box–whisker plots, where the whiskers in box plots show the minimum and maximum values with a line at the median. **(J)** (Statistical tests: (C, E, F) Student's unpaired *t* test. *P*-value < 0.05; **P*-value < 0.01; ***P*-value < 0.001; ****P*-value < 0.0001 and (J) one-way ANOVA with post hoc Tukey's multiple pairwise comparisons. *P*-value < 0.05; **P*-value < 0.01; ***P*-value < 0.001; ****P*-value < 0.0001).

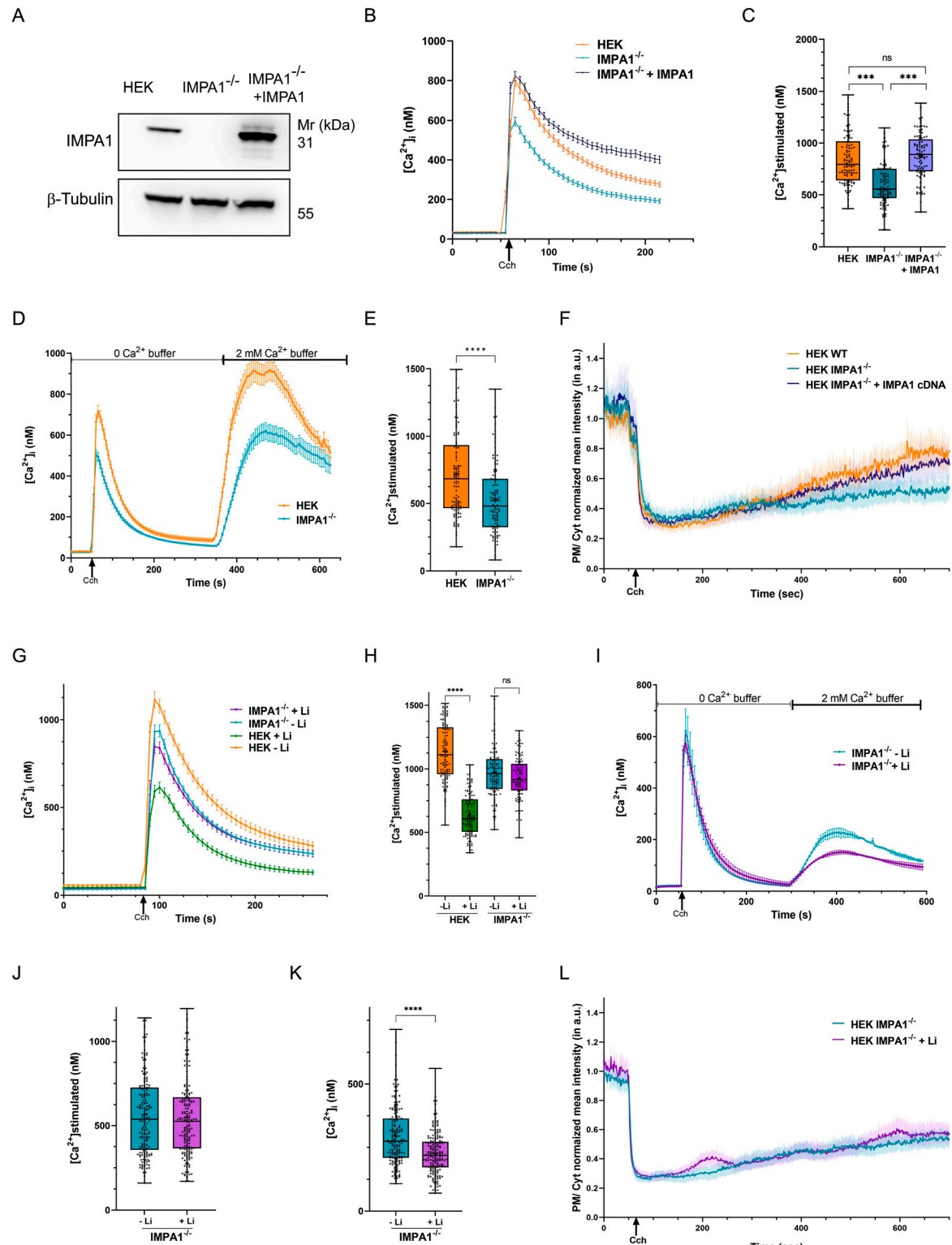

**Figure 3. Inositol monophosphatase (IMPA1) is required for the effect of Li on phosphatidylinositol 4,5-bisphosphate (PIP$_2$) resynthesis and agonist-mediated Ca$^{2+}$ signalling.**
**(A)** Western blot showing IMPA1 expression in IMPA1$^{-/-}$ cells and IMPA1$^{-/-}$ cells transduced by *IMPA1* cDNA. **(B, C)** PLC-dependent calcium mobilization (post stimulation with 20 µM of Cch) is decreased in IMPA1$^{-/-}$ cells; *IMPA1* expression via lentiviral transduction of cDNA in HEK IMPA1$^{-/-}$ cells rescued the decreased intracellular Ca$^{2+}$ release. Quantification: untreated HEK cells (orange), n = 155; HEK IMPA1$^{-/-}$ cells (light blue), n = 143; HEK IMPA1$^{-/-}$ cells + IMPA1 cDNA (dark blue), n = 162. **(D, E)** In zero Ca$^{2+}$ buffer, agonist-dependent calcium mobilization is decreased in HEK IMPA1$^{-/-}$ cells, compared with HEK cells. Quantification: HEK cells (orange), n = 121; HEK IMPA1$^{-/-}$ (blue),

(Table S1-Sheet 1 and Sheet 2). Gene Ontology (GO) analysis of the KEGG pathway showed substantial enrichment for genes annotated as being involved in calcium signalling or in glutamatergic synapse (Table S1-Sheet 3). In addition, GO KEGG analysis of the down-regulated gene set also showed enrichment for several other categories of neurotransmission and neuromodulation-related pathways (cholinergic, GABAergic, dopaminergic, glutamatergic, and serotonergic) (Table S1-Sheet 2) (Fig 6B).

Because GO analysis revealed an enrichment in $Ca^{2+}$ signalling genes, we analysed molecular mechanisms through which Li might down-regulate neuronal $Ca^{2+}$ signalling, and hence, excitability. There are multiple molecular mechanisms by which neuronal calcium signalling is modulated (Berridge, 1998) including voltage-gated $Ca^{2+}$ channels (VGCC) and receptor-activated $Ca^{2+}$ signalling, a mechanism used by several neurotransmitters and neuro-modulators. Using the transcriptomic data, we tested the likely contribution of each of these mechanisms to the effects of Li in our model system. Physiological and pharmacological evidence suggests that the $[Ca^{2+}]_i$ transients seen during neuronal development arise from the activity of VGCC (Rosenberg & Spitzer, 2011) and this also seems to underlie the $[Ca^{2+}]_i$ transients seen in our experiments when human iPSCs are differentiated into cortical neurons (Sharma et al, 2020). In this study, we find that the $Ca^{2+}$ transients monitored are also abolished by nimodipine and also by the application of tetradtoxin, a sodium channel blocker that alters membrane potential (Fig 4D and E), underscoring the role of VGCC activity in their origin.

To understand the mechanism by which Li could reduce the frequency of $[Ca^{2+}]_i$ transients, we studied the expression of genes encoding VGCC. Our transcriptomic analysis suggested that the expression of genes encoding several subunits of VGCC might be down-regulated (Table S1-Sheet 3). We experimentally tested the expression of genes for *CACNA1A*, *CACNA1B*, *CACNA1C*, *CAC1A1D*, and *CACN1A1E* by qRT–PCR analysis; this showed that the expression of CACNA1A, B, C, D were all down-regulated in Li-treated DIV45 neurons compared with controls (Fig 6C), whereas CACNA1E was unchanged. In the same analysis, there was no significant down-regulation in transcripts for *SCN1A (Nav1.1)* or *SCN9A (Nav7.1)* that encode the brain-enriched, α-subunit of voltage-gated sodium channels, involved in plasma membrane depolarization that trigger VGCC-dependent $[Ca^{2+}]_i$ transients (Fig 6C).

Glutamate is a key excitatory neurotransmitter in the human central nervous system, exerting its effects through both metabotropic and ionotropic mechanisms (Reiner & Levitz, 2018) and changes in $[Ca^{2+}]_i$ is a key mechanism in glutamate signalling in the human brain. Our qRT–PCR analysis of genes encoding key

components in the glutamate signalling system in Li-treated neurons revealed that transcript levels for metabotropic glutamate receptors (*GRM1*, *GRM3*, *GRM5*, *GRM7*) and the ionotropic glutamate receptors (*GRIA1*, *GRIA2*, *GRIN1*, *GRIN2A*, *GRIN2B*) were down-regulated (Fig 6D).

Many neurotransmitters and modulators transduce signal via G-protein-coupled, PLC-mediated store-operated $Ca^{2+}$ influx (Moccia et al, 2015). Key molecular components of this pathway include the $IP_3$ receptor, ER $Ca^{2+}$ sensor STIM, and the Ca influx channel ORAI; we found that the transcript levels of *PLCb4*, *ITPR2*, *STIM1*, and *ORAI2* were all down-regulated in Li-treated neurons compared with controls (Fig 6E).

## Li and GSK-3β inhibition induce distinct transcriptional responses

We also examined the transcriptional response of human iPSC derived forebrain cortical neurons to treatment with 10 μM CHIR99021 for 24 h. RNA-seq analysis revealed large changes in gene expression with ca. 5,000 genes each being up- or down-regulated (Fig S5C). GO analysis revealed a broad range of pathways including cardiomyopathy, amino acid metabolism, and extracellular matrix interactions (Fig 6F). Strikingly, there was no enrichment of pathways related to synaptic transmission or calcium signalling; the only brain-related pathway that was enriched was axon guidance and the identity of the genes picked up suggested ECM function. We overlapped the set of genes up- or down-regulated in Li and GSK-3β inhibition transcriptomes to assess the extent of overlap between the two gene sets and found that, both in the domain of up- and down-regulated genes (Figs 6G and H and S5D and E), there was at best a 3% overlap in terms of altered genes. These findings suggest that at the level of transcriptional changes, the mechanism of action of Li and GSK-3β is distinctive.

## Mechanisms of Li-induced transcriptional changes

To understand the mechanisms underlying Li-induced transcriptome changes, we analysed the likely transcriptional control mechanisms. For this, we used the htFtarget database that has information on all human transcription factors (TFs) and the genes they control, tissue-specific TF-target information, data on TFs for noncoding RNA, and information on co-regulation for a target gene and its TFs (Zhang et al, 2020). TFs controlling genes differentially expressed in Li-treated neurons (DIV45) were identified from the htFtarget database. Out of 650 up-regulated genes in Li-treated neurons, 587 were identified as associated with one or more TFs in this database. Likewise, out of 525 down-regulated genes, 333 genes

n = 120. **(F)** $PIP_2$ turnover post PLC stimulation in HEK cells compared with HEK IMPA1$^{-/-}$ cells; *IMPA1* expression via lentiviral transduction of cDNA in HEK IMPA1$^{-/-}$ cells rescued the delay in $PIP_2$ turnover. Mean ± 95% C.I. is plotted from three experiments, each performed in replicates. Quantification: untreated HEK cells (orange), n = 51; HEK IMPA1$^{-/-}$ cells (light blue), n = 50; HEK IMPA1$^{-/-}$ cells + IMPA1 cDNA (dark blue), n = 44. **(G, H)** Cch-mediated calcium release in these IMPA1$^{-/-}$ cells is not altered by lithium. Quantification: untreated HEK cells (orange), n = 126; HEK cells treated with Li (green), n = 127, HEK IMPA1$^{-/-}$ cells without Li treatment (blue), n = 132; HEK IMPA1$^{-/-}$ cells treated with Li (purple), n = 112. **(I, J, K)** In zero $Ca^{2+}$ buffer, agonist-dependent calcium mobilization is unaltered, but subsequent Store Operated Calcium Entry is decreased in lithium-treated IMPA1$^{-/-}$ cells. Quantification: HEK IMPA1$^{-/-}$ cells without Li treatment (blue), n = 179; HEK IMPA1$^{-/-}$ cells treated with Li (purple), n = 193. **(L)** Lithium does not alter the rate of regeneration of $PIP_2$ at the plasma membrane in the IMPA1$^{-/-}$ cells, as monitored by the changes in the translocation of the PH-PLCδ::GFP probe. Mean ± 95% C.I. is plotted from four experiments, each performed in replicates (HEK IMPA1$^{-/-}$ cells without Li treatment—blue line, n = 50; HEK IMPA1$^{-/-}$ cells treated with Li—purple line, n = 47). **(E, J, K)** (Statistical tests: (C, H) one-way ANOVA with post hoc Tukey's multiple pairwise comparison. *$P$-value < 0.05; **$P$-value < 0.01; ***$P$-value < 0.001; ****$P$-value < 0.0001 and (E, J, K) student's unpaired $t$ test. *$P$-value < 0.05; **$P$-value < 0.01; ***$P$-value < 0.001; ****$P$-value < 0.0001).

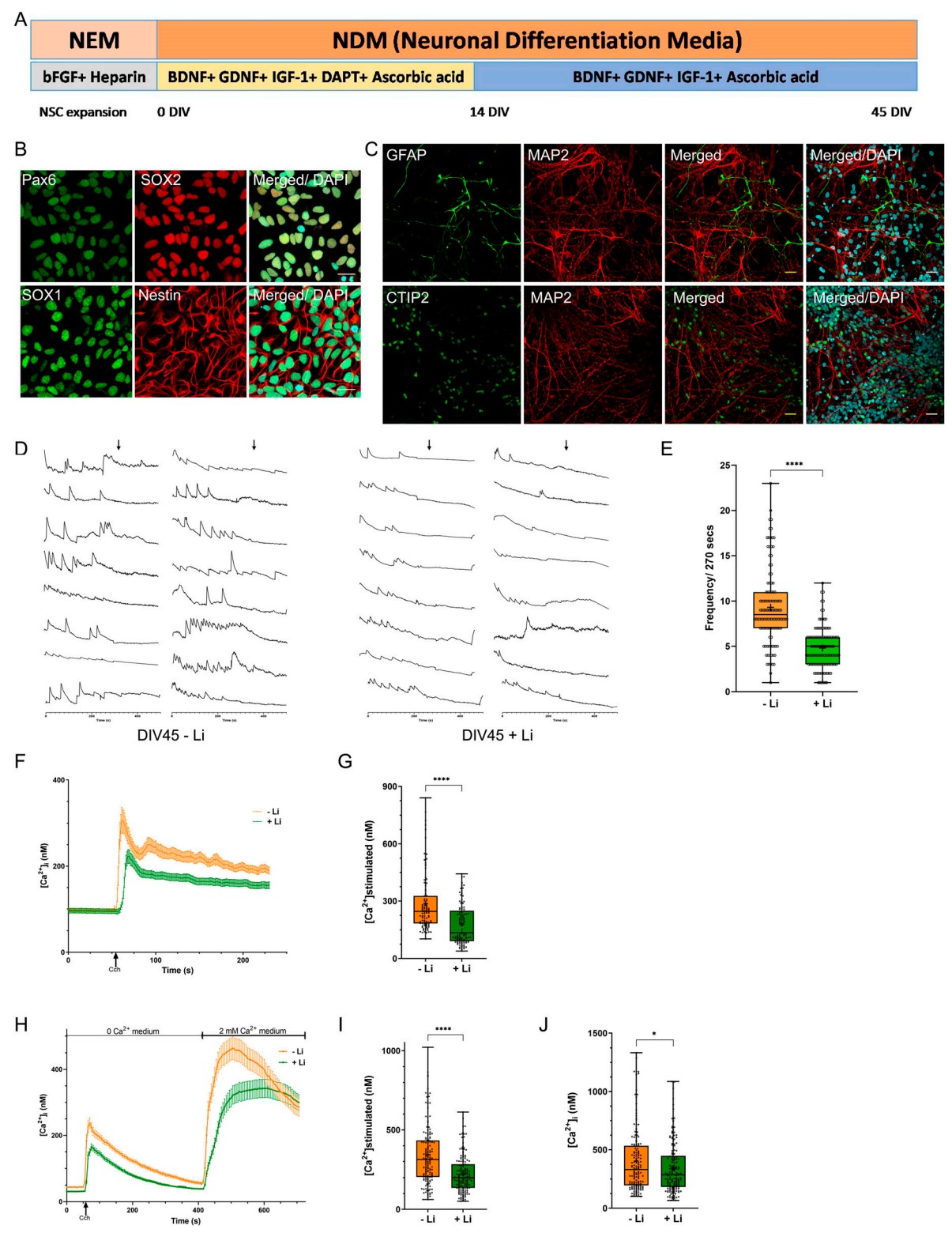

**Figure 4. Lithium reduces excitability in human cortical neurons.**
**(A)** Overview of expansion and differentiation process of the D149 lineage neural stem cells and the cortical neurons. **(B)** Characterization of neural stem cells via immunocytochemistry of known markers like Nestin, Sox2, Sox1, and Pax6. **(C)** The mature cortical neurons are characterized via immunocytochemistry of markers like MAP2, CTIP2, and GFAP (astrocyte marker) (Scale bar—25 $\mu$M). **(D)** $[Ca^{2+}]_i$ traces from individual cells soma; Y-axis shows $\Delta$F and X-axis is time in seconds. The baseline was recorded for 270 s followed by addition of 10 $\mu$M tetrodotoxin (as indicated by the arrows ↓) to block $Ca^{2+}$ transients. Transients are higher in frequency in the neuronal cultures without lithium treatment, compared with the treated cultures. The number of spikes/270 s are counted from individual soma and plotted, each dot representing

were linked to a TF. 485 and 323 unique TFs were identified that control the up-regulated and down-regulated genes, respectively. We then further looked for transcription factors that control differentially expressed genes that are involved in calcium signalling (Table S1-Sheet 3) (Fig S5A). Of the up-regulated genes, only RYR receptor was associated with three different transcription factors (CTCF, SPI1, and USF1) that could control its expression. On the other hand, several genes involved in calcium signalling that were down-regulated were controlled by transcription factors. Expression of *CACNA1C*, *CACNA1D*, *CAMK2B*, *CAMK4*, *CHRM3*, *GRIN1*, *ITPR2*, *RYR3*, *SCN9A*, and *SLC2A13* genes is regulated by transcription factors. Of these *ITPR2* and *CAMK2B* are regulated by 10 different TFs, whereas *CACNA1D*, *CHRM3*, *RYR3*, *SCN9A* and *SLC2A13* are regulated by a single transcription factor (Fig S5B). This suggests that some of the genes that are differentially expressed in Li-treated neurons are under transcriptional control which can be a direct or indirect effect of Li treatment.

## Discussion

Although Li is a monovalent ion with a remarkable therapeutic effect in the management of BPAD, there has been a lack of clear understanding on the mechanisms of action of Li in brain cells. Although many molecular targets of Li have been described in the literature, their role in mediating the action of Li in human brain cells has not been directly tested.

One of the earliest cellular effects described for Li was its ability to slow phosphoinositide turnover in neural cells (Berridge et al, 1982) and it has been proposed that the consequent slowing of neurotransmitter activated neuronal excitability may underlie the effectiveness of the compound in managing BPAD. Because Li inhibits the enzyme IMPA1, one long-standing proposal is that the application of Li to cells disrupts the receptor-activated $PIP_2$ cycle and its downstream effect on $[Ca^{2+}]_i$ signalling leading to reduced neuronal excitability. However, there has not been a direct test of the model that the ability of Li to modulate inositol turnover and $[Ca^{2+}]_i$ signalling in vivo requires the *IMPA1* gene product. In this study, we found that the application of Li to human cells reduced $[Ca^{2+}]_i$ signalling after PLC activation. We find that this is underpinned by a reduction in the release of $Ca^{2+}$ from intracellular stores. During PLC activation, the principal mechanism by which $Ca^{2+}$ is released is via the $IP_3$ receptor; however, we found that neither the transcript levels for this gene (Fig S2C) nor the size of stores itself was altered by Li treatment (Fig 2G and H). We also found that Li treatment did not alter the transcript levels of STIM and ORAI (Fig S2A and B). Thus, it seems most likely that Li treatment affects $IP_3$ receptor mediated $Ca^{2+}$ release by altering $IP_3$ production during receptor-mediated PLC stimulation, a process that requires

adequate levels of the substrate for PLC, $PIP_2$. Our finding that Li treatment slows synthesis of $PIP_2$, the substrate from which PLC produces $IP_3$, likely provides an explanation for the reduced intracellular $Ca^{2+}$ release phenotype seen in Li-treated cells after PLC stimulation.

Repeated cycles of GPCR-linked PLC signalling depend on a continuous supply of $PIP_2$ at the plasma membrane, whose synthesis in turn depends on the availability of inositol. Inositol levels in cells can be maintained by three avenues: (i) via the recycling through the stepwise dephosphorylation of $IP_3$, (ii) de novo synthesis from glucose 6-phosphate, and (iii) transport of inositol from the extracellular medium across the plasma membrane. Through its inhibition of IMPA1, Li can impact the supply of inositol via $IP_3$ dephosphorylation and de novo synthesis. However, the supply of inositol from the extracellular medium by sodium-dependent myo-inositol co-transporter and/or HMIT (proton-dependent myo-inositol co-transporter) is not likely to be impacted by Li treatment. We found that supplementation of the extracellular medium with enhanced levels of inositol could rescue the impact of Li application on $PIP_2$ resynthesis and $[Ca^{2+}]_i$ signalling. This finding implies that a reduced supply of inositol is key to the impact of Li on $PIP_2$ synthesis. $PIP_2$ levels are not affected under resting conditions in Li-treated cells, presumably because the low levels of inositol in the culture medium are sufficient to bypass the restriction in inositol supply from $IP_3$ dephosphorylation or glucose 6 phosphate to support PI and $PIP_2$ syntheses. However, during the intense PLC activity after receptor activation by agonists, the levels of inositol in the extracellular medium are insufficient to support inositol resynthesis leading to $PIP_2$ depletion. Our finding that inositol supplementation in the extracellular medium can rescue the reduced rate of $PIP_2$ resynthesis in Li-treated cells (Fig S2D) and the decreased PLC mediated $Ca^{2+}$ release phenotype (Fig 2I and J) supports this model.

If Li exerts its effects in neurons via inhibition of IMPA1, a prediction is that cells lacking IMPA1 might be unresponsive to Li. We generated a cell line in which IMPA1 was deleted and found that in the absence of IMPA1, Li was unable to exert its effects on both $[Ca^{2+}]_i$ signalling and $PIP_2$ resynthesis during receptor-activated PLC signalling. These findings imply that an intact IMPA1 is required for the action of Li on these processes. Consistent with this idea, we noted that receptor-activated $[Ca^{2+}]_i$ signalling (Fig 3D and E) and the rate of $PIP_2$ resynthesis (Fig 3F) after PLC activation was reduced in IMPA1$^{-/-}$ cells without Li treatment. Therefore, this study provides compelling evidence for the link between the inhibition of IMPA1 by Li leading to reduced $PIP_2$ synthesis and thence to reduced neurotransmitter-activated, PLC-mediated $Ca^{2+}$ signalling. Although we generated IMPA1$^{-/-}$ NSC, we were unable to differentiate these into cortical neurons as NSC of this genotype undergo cell death soon after the initiation of differentiation. This finding is consistent with a previous report that stem cell lines from

---

events from a single soma. **(E)** Data is plotted from multiple experiments, each done with biological replicates (untreated neurons (orange), n = 114; neurons treated with Li (green), n = 113). **(F, G)** Cch-dependent calcium mobilization was decreased in lithium-treated neuronal cultures (DIV45). Quantification: untreated neurons (orange), n = 97; neurons treated with Li (green), n = 106. **(H, I, J)** Decrease in the Cch-dependent calcium mobilization and in store-operated calcium entry cells in lithium treated neuronal cultures (DIV45). Quantification: untreated neurons (orange), n = 154; neurons treated with Li (green), n = 156. (Statistical tests: (E, G, I, J) Mann–Whitney test. *$P$-value < 0.05; **$P$-value < 0.01; ***$P$-value < 0.001; ****$P$-value < 0.0001).

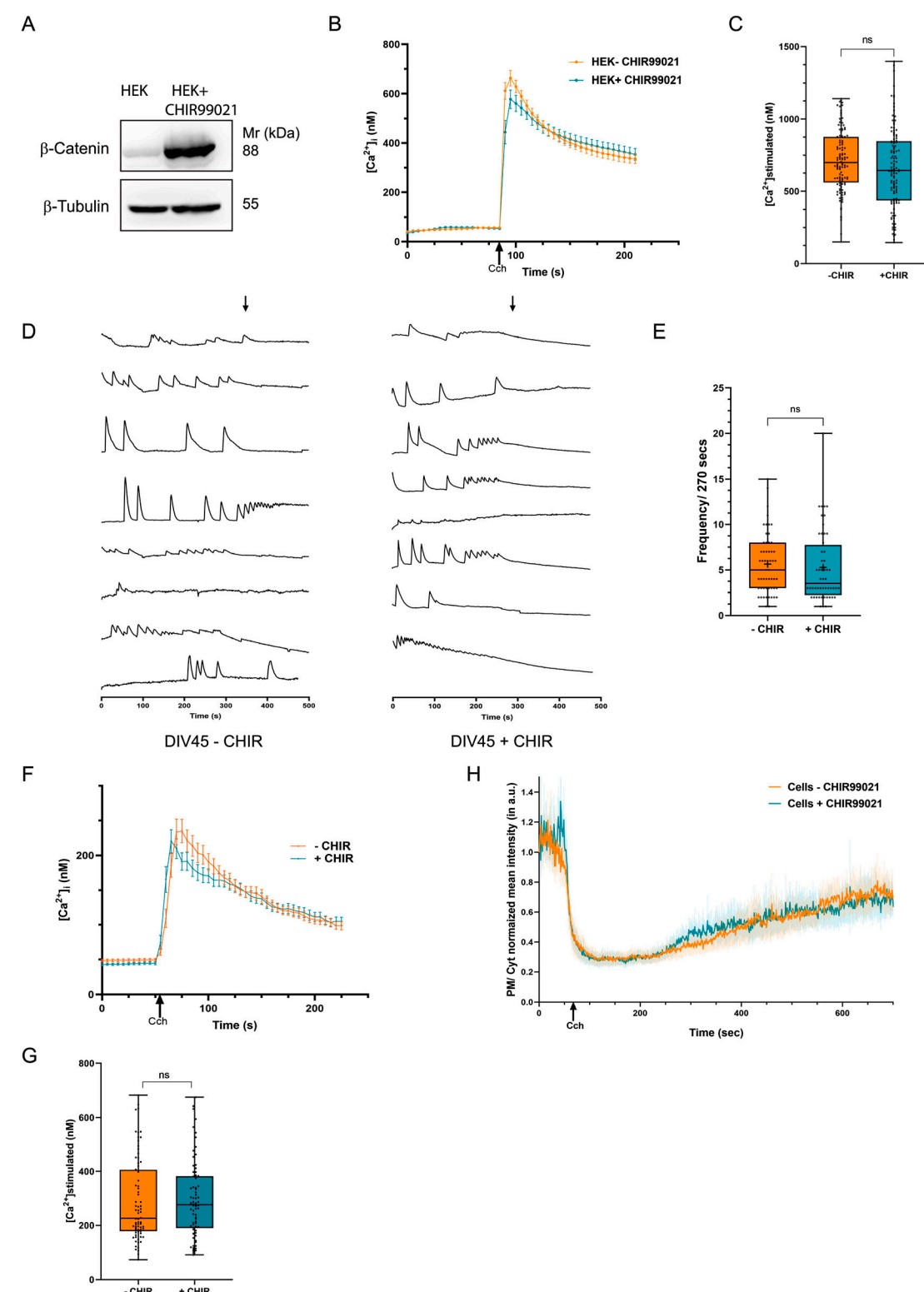

**Figure 5. Glycogen synthasekinase-3β (GSK-3β) inhibition by CHIR99021 does not affect PLC mediated Ca²⁺ mobilization or phosphatidylinositol 4,5-bisphosphate turnover.**
**(A)** Western blot showing CHIR99021 (an inhibitor of GSK-3β) treatment led to the high expression of β-catenin, one of the targets of GSK-3β destruction complex.
**(B, C)** Inhibition of GSK-3β by CHIR99021 did not alter agonist-mediated Ca²⁺ release. Quantification: untreated cells (orange), n = 116; cells treated with CHIR99021 (blue), n = 118. **(D)** $[Ca^{2+}]_i$ traces from individual cells (DIV45 neurons) untreated or treated with CHIR99021 (soma; Y-axis shows ΔF and X-axis is time in seconds). The number of spikes/270 s are counted from individual soma and plotted, each dot representing events from a single soma. **(E)** Quantification- untreated neurons (orange), n = 55;

a human patient with an IMPA1 mutation could not be differentiated into cortical neurons (Figueiredo et al, 2021).

How does IMPA1 regulate $PIP_2$ synthesis and $[Ca^{2+}]_i$ signalling? IMPA1 dephosphorylates inositol 1-phosphate to generate inositol. Hence, biological effects resulting from Li inhibition of IMPA1 could arise either from an accumulation of the substrate, inositol 1-phosphate or a deficiency of the product, inositol. Indeed, Li treatment in rodent models has shown both an elevation of inositol 1-phosphate levels and a modest reduction in inositol levels (Sherman et al, 1981; Sade et al, 2016). In our analysis, we found that the lowered $[Ca^{2+}]_i$ influx after receptor activation in human cells could be rescued by supplementation of the extracellular medium with inositol. This observation implies that the $[Ca^{2+}]_i$ signalling defect arising from Li inhibition of IMPA1 is likely to be a consequence of inositol depletion rather than an accumulation of the substrate inositol 1-phosphate. During the PLC-activated $PIP_2$ cycle, inositol generated by the action of IMPA1 is condensed with cytidine diphosphate DAG to form phosphatidylinositol (Lykidis et al, 1997), which is then sequentially phosphorylated to generate $PIP_2$. In this study, we found that after PLC activation, the resynthesis of $PIP_2$ was also slowed by treatment of WT cells with Li. This observation is consistent with our finding that $PIP_2$ resynthesis and $[Ca^{2+}]_i$ signalling were both reduced in IMPA1$^{-/-}$ cells. Together, our data provide compelling evidence that Li treatment results in inhibition of IMPA1, a depletion of the inositol pool required for $PIP_2$ resynthesis during PLC signalling and hence reduced activity of neurons during neurotransmitter-activated synaptic transmission.

An alternative and widely discussed molecular target of Li is GSK-3$\beta$. However, in this study, we found that in the hiPSC-derived cortical neuron cultures, inhibition of GSK-3$\beta$ did not phenocopy the physiological effects of Li treatment on neuronal excitability and PLC signalling. Furthermore, RNA seq analysis revealed minimal overlap in the transcriptome changes induced by Li treatment and GSK-3$\beta$ inhibition. Thus, it seems, in the hiPSC model system, Li is unlikely to influence neuronal excitability via GSK-3$\beta$ inhibition.

When Li is used to treat BPAD in human patients, presumably, it acts by reducing the excitability of neurons in the cerebral cortex. In this study, we tested the effect of Li treatment on the physiology of human forebrain cortical neuronal cultures differentiated from iPSC. Consistent with previous reports (Mertens et al, 2015), we found that treatment of human forebrain cortical neurons with therapeutically relevant concentrations of Li reduced the frequency of $[Ca^{2+}]_i$ transients that are underpinned by VGCC activity (Sharma et al, 2020). Because acute application of Li does not affect $[Ca^{2+}]_i$ transients (Fig S4D and E), it seems unlikely that Li affects neuronal excitability by directly inhibiting the VGCC. However, VGCC-mediated $[Ca^{2+}]_i$ transients are activated by neuronal action potentials, which themselves are triggered after activation of postsynaptic G-protein-coupled receptors that bind neurotransmitters. Several of these receptors (mGluR1, mGluR5, mAchR, and 5HT-2A) when bound to

their respective neurotransmitter ligand, use PLC activation as part of their signalling mechanism. Thus, Li could influence neuronal excitability by modulating signalling through such PLC-linked receptors. In support of this hypothesis, in this study, we found that in human forebrain cortical neurons, in addition to reducing $[Ca^{2+}]_i$ transients, Li application diminished the elevations of $[Ca^{2+}]_i$ triggered by application of the mAchR ligand carbachol (Fig 4F–I). Thus, a key mechanism by which Li modulates neuronal excitability may be through down-regulation of neurotransmitter signalling of PLC-linked GPCRs.

Given the extended timeframe over which Li exerts its therapeutic effect, it has been proposed that in addition to directly impacting cellular biochemistry, transcriptional mechanisms may play a role in its biological effects. SOCE that occurs downstream of receptor-activated PLC signalling has been shown to regulate transcription in neurons (reviewed in Mitra and Hasan [2022]). Our transcriptomic analysis of human forebrain cortical neurons revealed that Li treatment on a clinically relevant time and concentration range results in a large transcriptional response. Likewise, a comparison of Li-induced differential gene expression (this study) to that elicited by genetic inhibition of SOCE (Gopurappilly et al, 2018; Dhanya & Hasan, 2021) also revealed only a modest overlap of genes that were differentially regulated (Fig S4F and G). Therefore, it is likely that Li exerts transcriptional changes in neurons independent of its effects on SOCE.

Our analysis of the Li-induced transcriptome changes in human forebrain cortical neurons however provides an insight into the mechanisms by which Li might mediate its effects both in cultured forebrain cortical neurons and its clinical effects. We found that the levels of three of the four subunits of VGCC were down-regulated in Li-treated cells and this could explain, in part, the ability of Li to reduce $[Ca^{2+}]_i$ transients in forebrain cortical neurons in culture. A key feature of mania in BPAD patients is heightened neural activity manifested behaviorally, EEG and fMRI studies and presumably Li exerts its therapeutic effect through down-regulating this neural activity. Glutamate is the principle excitatory neurotransmitter in the brain and, consistent with this, we found that transcripts for multiple glutamate receptor subtypes were down-regulated in forebrain cortical neurons treated with Li. These included multiple subunits of the metabotropic, AMPA, and NMDA subtypes of the glutamate receptor family. Down-regulation of these will likely contribute to reducing neural activity after Li treatment manifest clinically as behavioral improvement. Overall, this study demonstrates that the effects of Li on neuronal $Ca^{2+}$ signalling are mediated through an IMPA-dependent step and involves a large transcriptional response that down-regulates molecular processes that are required for neuronal excitability. The link between Li and transcriptional responses is unclear although studies in yeast have shown that treatment with Li is associated with changes in the transcriptome that also includes changes in inositol biosynthesis.

---

neurons treated with CHIR99021 (blue), n = 52. **(F, G)** Cch dependent calcium mobilization was unchanged in CHIR99021-treated neuronal cultures (DIV45). Quantification: untreated neurons (orange), n = 76; neurons treated with CHIR99021 (blue), n = 74. **(H)** Phosphatidylinositol 4,5-bisphosphate turnover post PLC stimulation in HEK cells treated with CHIR99021, compared with untreated cells (untreated cells- orange line, n = 31, cells treated with CHIR99021- blue line, n = 34). (Statistical tests: (B, C) student's unpaired $t$ test. *$P$-value < 0.05; **$P$-value < 0.01 ***$P$-value < 0.001; ****$P$-value < 0.0001).

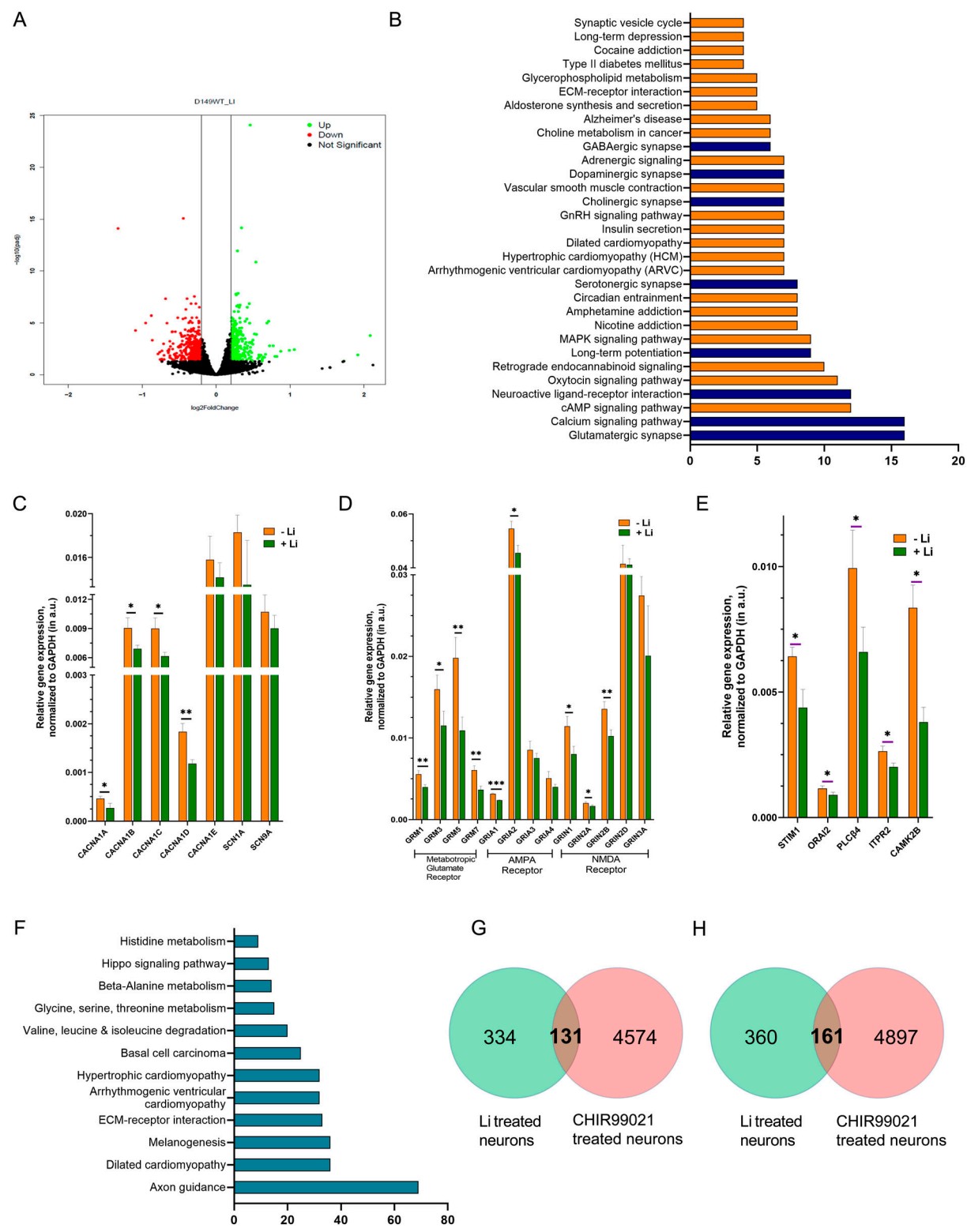

**Figure 6. Lithium treatment induces transcriptional changes in pathways involved in neuronal transmission.**
**(A)** Volcano plot showing the expression of genes altered because of lithium treatment in human cortical neuronal cultures DIV45 (P-value < 0.05, P-adj value < 0.01). **(B)** The GO KEGG pathway enrichment analysis for down-regulated genes. **(C, D, E)** Quantitative analysis showing several genes related to voltage calcium channels (C), Glutamatergic pathway (D) and calcium homeostasis (E), are down-regulated in cortical neurons (DIV45) because of lithium treatment. **(F)** The GO KEGG pathway enrichment analysis for down-regulated genes because of CHIR99021 treatment. **(G, H)** Distribution of the number of up-regulated and down-regulated genes in neuronal cultures (DIV45) because of Li treatment and CHIR99021 treatment. (Statistical tests: (C, D, E) Multiple unpaired t test. *P-value < 0.05; **P-value < 0.01; ***P-value < 0.001.)

# Materials and Methods

### DNA constructs

For PIP$_2$ measurement, PH-PLC$\delta$::GFP probe was used, which was a gift from Tamas Balla (plasmid # 51407; Addgene). The cDNA for human muscarinic acetylcholine receptor m1AchR was obtained from cDNA Resource Center (MAR0100000) and was cloned via Gibson assembly into a lentiviral pHR transfer backbone (having Puromycin resistance cassette) for lentiviral generation. The Nir2 shRNA was obtained from GE Dharmacon (RHS4430-200186441: Clone id V2LHS_62836).

### Mammalian cell culture, treatment, and transfection

HEK293T cells were maintained under standard conditions at (37°C, with 5% CO$_2$) in DMEM High Glucose (DMEM; Life Technologies), supplemented with 10% FBS. For Li treatment, the cells were grown and maintained in these media, supplemented with 1 mM LiCl (Sigma-Aldrich) for 5 d (unless mentioned otherwise). The cells were tested for mycoplasma, before the experiments. MycoAlert (Lonza) was used per the manufacturer's protocol to test the spent media for mycoplasma contamination. Transfection with the plasmid expressing PH-PLC$\delta$::GFP probe was done 24 h before the imaging experiments using polyethylenimine solution (1 mg/ml; PEI; Polysciences Inc.) when the cells were around 70% confluent.

### Lentiviral transduction to generate HEK293T cell line stably expressing muscarinic acetylcholine receptor m1AchR

Recombinant lentiviral particles were packaged in HEK293T cells grown in DMEM High Glucose media, supplemented with 10% FBS. The HEK293T cells cultured in the six-well plates were transfected with a solution consisting of 1.2 $\mu$g of pHR-IRES plasmid encoding muscarinic acetylcholine receptor *CHRM1* cDNA, along with 1.2 $\mu$g of each of the helper plasmids (psPAX and pMD2.G) in 500 $\mu$l of Opti-MEM (Invitrogen) and 110 $\mu$l polyethylenimine (1 mg/ml; PEI). The medium was changed after 12 h and the virus was harvested at 72 h after transfection. Then, HEK293T cells were transduced with these recombinant viral particles and selected via puromycin treatment (1.5 $\mu$g/ml) for 1 wk to obtain the HEK293T cell line, expressing m1AchR. Western blot confirming the expression of m1AchR is shown in Fig S3C and D.

### Generation of NSCs and differentiation and maintenance of neuronal culture

NSCs were generated as previously described (Mukherjee et al, 2019), with slight modifications. Human iPSCs were differentiated to embryoid bodies in E6 medium (Gibco). After that, primary neural rosettes that were formed, were manually picked, passaged, and eventually plated as NSC monolayer in Neural Expansion Medium (NEM; DMEM/F12 with 2 mM Glutamax, supplemented with N2, B27 without vitamin A, 20 ng/ml bFGF, 100 $\mu$M nonessential amino acids, 2 $\mu$g/ml heparin sulphate, 100 U/ml penicillin–streptomycin) (all components procured from Life Technologies). The NSCs were

subjected to CD133+ selection as previously described via FACS (Peh et al, 2009) to eliminate cells of other lineages. The sorted NSCs were further characterized by immunofluorescence for the specific markers. NSCs were grown on Matrigel-coated tissue dishes in NEM at 37°C and 5% CO$_2$.

For differentiation, NSCs were seeded onto Matrigel- Growth Factor Reduced (Corning) coated cover-slip dishes or 35 mm dishes and expanded in NEM till the cells reached 70% confluency. NEM was then withdrawn and replaced with Neural Differentiation Medium (NDM; DMEM/F12 with 2 mM Glutamax, supplemented with N2, B27 containing vitamin A, 100 $\mu$M nonessential amino acids, 100 U/ml penicillin–streptomycin, 10 ng/ml BDNF, 10 ng/ml GDNF, 10 ng/ml IGF-1, 50 $\mu$g/ml ascorbic acid [Sigma-Aldrich] and 2 $\mu$M DAPT [Sigma-Aldrich]). DAPT was supplemented for the first 14 d. Media were changed every 2 d during the differentiation process. All the experiments were done DIV45 post differentiation (The day of addition of NDM was DIV0). For chronic Li treatment, the neuronal cultures were maintained in NDM supplemented with 1 mM LiCl for the last 10 d (DIV35–DIV45).

### Live cell imaging for the PIP$_2$ probe

To analyse the dynamics of PIP$_2$ at the plasma membrane by following the change in translocation of the PH-PLC$\delta$::GFP probe, live cell imaging was performed as stated in the study by Várnai and Balla (1998), with minor modifications. 20,000 cells were seeded in Poly-L Ornithine -coated cover-slip dishes and grown for 96 h with or without Li. 24 h before imaging, the cells were transfected with PH-PLC$\delta$::GFP plasmid using polyethylenimine as per the manufacturer's protocols. Carbamoylcholine chloride mediated changes in the membrane localization of PH-PLC$\delta$::GFP probe was examined by a time-lapse method using an Olympus FV 3000 confocal microscope. The cells were kept in pre-warmed Kreb's Ringer Buffer (120 mM NaCl, 4.7 mM KCl, 1.2 mM CaCl$_2$, 0.7 mM MgSO$_4$, 10 mM glucose, 10 mM HEPES; pH-7.4) and at 37°C while imaging. Changes in the PIP$_2$ membrane localization were quantified by taking a fluorescence intensity ratio of the plasma membrane (F$_{PM}$) and the cytosol (F$_{Cyt}$) across the time-lapse and normalizing it to the first value.

### Calcium imaging for carbamoylcholine stimulation

For studying the calcium physiology, HEK293T-M1 cells were seeded in Poly-L Ornithine coated cover-slip dishes, whereas iPSC-derived cortical neurons (DIV45) were grown on Matrigel-coated cover-slip dishes. Before imaging, the cells were washed twice with HBSS (10 mM HEPES, 118 mM NaCl, 4.96 mM KCl, 1.18 mM MgSO$_4$, 1.18 mM KH$_2$PO$_4$, 10 mM Glucose, 2 mM CaCl$_2$; pH-7.4) and then loaded with the ratiometric Ca$^{2+}$ indicator Fura-2 AM (4 $\mu$M) (Acetoxymethylester; Invitrogen) along with 0.002% Pluronic F-127 for 30 min. The excess dye was removed by washing the cells thrice with HBSS and cells were incubated for an additional 20 min to equilibrate the intracellular dye concentration and allow de-esterification. Imaging was performed for 10 min using a 40X objective (N.A. 0.75) of wide-field fluorescence microscope Olympus IX-83 for HEK293T cells, whereas a 20X objective (N.A. 0.50) was used for neuronal cultures. Free- and Ca$^{2+}$-bound Fura-2 fluorescence intensities (excitation 340/380 nm, emission 510 nm) were recorded at every 5-s intervals using an

EM-CCD camera (Evolve 512 Delta; Photometrics). Basal cytosolic $Ca^{2+}$ was recorded in HBSS buffer for the initial 18 frames. Carbamoylcholine chloride (20 and 50 $\mu M$, respectively, for HEK and neuronal cultures, as determined in preliminary concentration–response experiments) (C4382; Sigma-Aldrich) was added to the cells for activating PLC and thereby, inducing $Ca^{2+}$ release from the endoplasmic reticulum store. After recording for 42 frames, 20 $\mu M$ Ionomycin (Calbiochem) was added and recorded for 24 frames for recording $R_{max}$. 24 more frames were recorded, post addition of 4 mM EGTA (containing 0.01% Triton-X-100) for chelation of $Ca^{2+}$ ions and determination of $R_{min}$.

**Calcium imaging for SOCE**

For the SOCE experiment, the cells were kept in "zero $Ca^{2+}$ HBSS" (10 mM HEPES, 118 mM NaCl, 4.96 mM KCl, 1.18 mM $MgSO_4$, 1.18 mM $KH_2PO_4$, 10 mM glucose; pH-7.4) just before the imaging. After recording the initial 12 frames for basal cytosolic $Ca^{2+}$, an optimal concentration of thapsigargin (Tg) (10 $\mu M$) (Invitrogen) was added for the store depletion and 60 frames were recorded. Then, 2 mM $CaCl_2$ was then added to the extracellular buffer to induce SOCE—60 frames were recorded post induction of SOCE. Subsequently, 20 $\mu M$ Ionomycin (Calbiochem) was added and recorded for 24 frames; then 4 mM EGTA (containing 0.01% Triton-X-100) was added and 24 frames were recorded. For studying PLC-activated calcium mobilization from stores and SOCE, carbamoylcholine chloride (20 $\mu M$ and 50 $\mu M$, respectively, for HEK and neuronal cultures) was added instead of thapsigargin.

**Analysis of calcium recordings**

The CellSens Olympus software was used for the analysis. A dynamic region of interest (ROI) was drawn for each cell which exhibited a rise in $[Ca^{2+}]_i$ (as seen by increase in fluorescence post-stimulation) to track the fluorescence changes over time. The emission intensities corresponding to 340- and 380-nm excitations were used to calculate the 340/380 ratio for each cell across all time points. The $[Ca^{2+}]_i$ was estimated using the Grynkiewicz equation (Grynkiewicz et al, 1985) as follows:

$$[Ca^{2+}]_i (nM) = K_d \times SF \times (R - R_{min}/R_{max} - R)$$

where $R_{min}$ and $R_{max}$ refer to the minimum and maximum 340/380 ratio, respectively. 224 nM was taken as $K_d$ for Fura-2 AM in human cells and SF (scaling factor) was calculated for each cell by dividing the fluorescence emission intensity at $Ca^{2+}$ free form with the fluorescence emission intensity at $Ca^{2+}$ bound form of the dye after excitation at 380 nm. The final $Ca^{2+}$ traces were plotted using GraphPad Prism 9.0.

**Generation of the IMPA1 deletion in cell lines**

For generation of HEK293T-IMPA1$^{-/-}$ cell line, the initial region of the coding sequence (exon1) of the IMPA1 gene was targeted by a pair of sgRNAs and deleted via CRISPR-Cas9 technology, then these cells were transduced to stably express Muscarinic receptor M1. The sgRNAs targeting the start of the coding sequence of IMPA1 gene

were designed via the crispr.mit.edu. The sequences of the sgRNAs are as follows:

5′ GGAGAACCACCTTGTTGGCCAAGCTGGAATGTAGTGGCGTTTTAGAGCTA-GAAATAGCAAGTT 3′

5′ GGAGAACCACCTTGTTGGGTAATATGGTACAGACACACGTTTTAGAGCTA-GAAATAGCAAGTT 3′

These sgRNAs were cloned individually into humanized pgRNA plasmids (pgRNA-humanized was a gift from Stanley Qi [plasmid # 44248; Addgene]). The cells were co-transfected with the three plasmids (both the pgRNA-humanized plasmids encoding the sgRNAs and pLentiCas9 blast encoding for the Cas9 protein endonuclease from *Streptococcus pyogenes*) and selected with puromycin (2 $\mu g$/ml) for 48 h. Then the genomic DNA was isolated, and a Surveyor PCR was performed to check the occurrence of the deletion. Once confirmed that the sgRNAs were able to target the exon 1 of IMPA1 gene, the transfected cells were seeded in 96 wells as single cells and gradually expanded. After selection via Surveyor PCR using Surveyor primers (primers flanking the region ought to be deleted), a HEK293T cell line was selected which had showed only the shorter amplicon (amplicon after deletion) in the Surveyor PCR. This HEK293T cell line was then expanded and the expression of IMPA1 was checked by Western blotting using the anti-IMPA1 antibody. No corresponding bands for IMPA1 were observed, indicating that IMPA1 was knocked out in this cell line. This HEK293T-IMPA1$^{-/-}$ cell line did not show any morphological difference with respect to HEK293T cells and did not require inositol supplementation in the media for maintenance.

**Western blotting**

For estimating the expression of the IMPA1, cells were suspended in appropriate volume of lysis buffer and kept on ice for lysis; 10 $\mu l$ of it was used for Bradford protein assay (Bio-Rad) for quantification. Post quantification, an equal amount of samples was heated at 95°C with Laemmli loading buffer for 5 min, then loaded and separated on 10% SDS–PAGE gel. For estimating the expression of the muscarinic acetylcholine receptor m1AchR, cells were pelleted and directly lysed by adding Laemmli's loading buffer and repeated syringing with insulin syringes (31 gauge). Then the samples were heated at 95°C for 5 min and loaded onto a pre-cast polyacrylamide 4–12% gradient gel (Bolt, Invitrogen). The proteins were then transferred onto a nitrocellulose membrane and then blocked with 10% Blotto (Santa Cruz Biotechnology) diluted in Tris Buffer Saline containing 0.1% Tween-20 (0.1% TBST) for 40 min. Subsequently, the membrane was incubated with the rabbit anti-IMPA1 antibody (1:10,000 diluted in 5% BSA in 0.1% TBST) (ab184165; Abcam) or goat anti-CHRM1 antibody (1 $\mu g$/ml, diluted in 5% BSA in 0.1% TBST) (ab77098; Abcam) overnight at 4°C. For loading control, the blot was incubated with mouse anti-$\beta$-tubulin antibody (1:4,000 diluted in 5% BSA in 0.1% TBST) (DSHB Hybridoma Product E7). The blots were then washed thrice with 0.1% TBST and then incubated with the appropriate HRP-conjugated secondary antibodies (1:10,000 dilutions; Jackson Laboratories, Inc.) for 1.5 h. Post incubation with the secondary antibodies, the blot was washed thrice with 0.1% TBST and developed using Clarity Western ECL substrate (Bio-Rad) on a GEImageQuant LAS 4000 system. The ImageJ software was used for the densitometric analysis of the blots. Firstly, the background intensities were subtracted from the images using an

average of mean background intensities of a few ROIs adjacent to the band of interest. Then ROIs were drawn around the bands of interest and the integrated density of each ROI was extracted from the image. Finally, the ratio of the integrated density of the IMPase to that of the loading control ($\beta$-Tubulin) was calculated.

## Calcium imaging for measuring transients

For studying the effect of Li on the excitability of cortical neurons, spontaneous calcium events (calcium transients) were measured in D149 cortical neuronal cultures at DIV45 for both untreated and Li-treated neurons, according to previously published lab protocols (Sharma et al, 2020). The neuronal cultures were washed twice for 10 min with HBSS and then loaded with 4 $\mu$M of Fluo-4 AM (F14201; Molecular probes) in HBSS having 0.002% Pluronic F-127 for 30 min at RT. Imaging was performed for a time-span of 10 min using a 20X objective (N.A. 0.50) with a time interval of 1 s at 488 nm illumination. The baseline measurement was recorded for a period of 4 min to visualize the calcium transients. Then 10 $\mu$M tetrodotoxin (Hellobio) was added to abolish calcium transients and recorded for another 4 min. A dynamic ROI was manually drawn around each neuronal soma; individual $Ca^{2+}$ traces were then obtained using the CellSens Dimensions software. The fluorescence intensity value from each neuronal soma was normalized to the first baseline signal and the individual traces were plotted using GraphPad Prism 9.0. The frequency of calcium transients was measured manually from first-derivative filter traces.

## Post-hoc immunofluorescence for neuronal characterization

Post calcium imaging, the neuronal cultures were washed thoroughly a few times with HBSS buffer and then fixed with 4% PFA dissolved in PBS for 20 min. Post fixation, the cultures were washed twice with PBS for 5 min each and then permeabilized with 0.1% Triton-X (dissolved in PBS). The cultures were then blocked with 5% BSA diluted in PBS, for 1 h at RT. Subsequently, the neurons were incubated with the primary antibodies diluted in 5% BSA in PBS overnight at 4°C. Next day, the cultures were subjected to three washes with PBS (10 min each) to remove nonspecifically bound primary antibody. Respective secondary antibodies at a dilution of 1:300 in 5% BSA in PBS were used for incubation for 2 h. DAPI was used as a nuclear marker at a dilution of 1:1,000. The primary and secondary antibodies used for immunocytochemistry are listed in Table S2.

## RNA isolation, cDNA synthesis and real-time quantitative PCR

The total RNA was extracted from HEK293T cells or mature neuronal cultures, in multiple biological replicates using TRIzol (Ambion, Life Technologies) following the manufacturer's protocol and thereby, quantified via NanoDrop 1,000 spectrophotometer (Thermo Fisher Scientific). For cDNA synthesis, 1 $\mu$g of the RNA from each replicate was treated with DNase I (amplification grade, Thermo Fisher Scientific) and then incubated with Superscript II reverse transcriptase (Invitrogen) along with random hexamers and dNTPs. A no reverse transcription control sample was also included for each sample. Real-time quantitative PCR analysis was performed on an Applied Biosystems 7500 fast qRT-PCR system using diluted cDNA samples and primers against genes of interest and GAPDH as internal controls. The $C_t$ values obtained for different genes were normalized to those of GAPDH from the same sample. The relative expression levels were calculated using $\Delta C_t$ method, whereas the fold change was calculated using $\Delta\Delta C_t$ method. The primers used for qRT–PCR are listed in the Table S3.

## LC-MS–based PIP and PIP$_2$ measurements

To validate whether the PH-PLC$\delta$-GFP probe-based imaging of PIP$_2$ in the HEK293T cells reflected the changes in PIP$_2$ level, mass spectrometry measurements of PIP$_2$ from the cells were carried out following existing laboratory protocols (Sharma et al, 2019). Reverse phase liquid chromatography was coupled with high-sensitivity mass spectrometry (LC-MS) and a multiple reaction monitoring (MRM) method was employed to detect PIP$_2$ levels from the whole cells.

### Lipid extraction

Equal number of cells from a single well of a 12-well plate (counted by a haemocytometer while seeding) were pelleted and then gently resuspended into 170 $\mu$l of 1X PBS in a 2 ml low-bind polypropylene centrifuge tube. To this, 750 $\mu$l of ice-cold quench mixture (MeOH/CHCl$_3$/1 M HCl in the volumetric ratio of 484/242/23.55), followed by 15 $\mu$l of a pre-mixed ISD (Internal Standard) mixture containing 25 ng of 37:4 PIP (**PI 4P 17:0/20:4 [LM-1901]**), 25 ng of 37:4 PIP$_2$ (**PI 4,5P$_2$ 17:0/20:4 [LM-1904]**), and 50 ng of 31:1 phosphatidylethanolamine (**PE 17:0/14:1 [LM-1104]**) was added (All the lipid standards were procured from Avanti Lipids). For carbamoylcholine chloride-mediated PLC activation, cells were suspended in 85 $\mu$l of 1XPBS and then 85 $\mu$l of 20 $\mu$M Carbamoylcholine chloride (made in PBS) (final concentration—10 $\mu$M) was added for 1 min at 37°C in shaking conditions. For the regeneration of PIP$_2$ post PLC-mediated hydrolysis, 1 ml of ice-cold PBS was added to dilute the agonist carbamoylcholine chloride and stop the stimulation. Cells were pelleted and washed and then resuspended in pre-warmed buffer (37°C) and kept for 15 min for recovery (time given for regeneration of PIP$_2$). Post 15 min of incubation, cells were pelleted and then resuspended in 170 $\mu$l of PBS, to which 750 $\mu$l of quench mixture and 15 $\mu$l of a pre-mixed ISD were added as stated previously. The mixture was vortexed and 725 $\mu$l of CHCl$_3$ and 170 $\mu$l of 2.4 M HCl were added. After vortexing again for 2 min at 1,500 rpm, the phases were separated by centrifugation for 3 min at 1,500$g$. The lower organic phase was then collected and added to fresh tubes containing 708 $\mu$l of Lower Phase Wash Solution (LPWS; MeOH/1 M HCl/CHCl$_3$ in the volumetric ratio of 235/245/15). The tubes were then vortexed and spun at 1,500 rpm again for 3 min to separate the phases. The resultant lower organic phase was collected into a fresh tube and subjected to the derivatization reaction.

### Derivatization of lipids

50 $\mu$l of 2 M TMS–diazomethane was added to the collected organic phase (TMS–diazomethane is toxic and is to be used with utmost precautions and following safety guidelines). The reaction was allowed for 10 min at 600 rpm at RT, post which, 10 $\mu$l of glacial acetic acid was added to quench the reaction. 500 $\mu$l of post derivatization wash solution (PDWS- CHCl$_3$/MeOH/H$_2$O in the volumetric ratio of 24/12/9; the upper phase was used) was then added to the sample, vortexed and spun down at 1,500$g$ for 3 min. The upper aqueous

phase was discarded, and the wash step was repeated. To the final organic phase, 45 $\mu l$ MeOH and 5 $\mu l$ $H_2O$ was added, mixed, and spun down. The samples were then dried in a SpeedVac at 400 rpm for 150 min under vacuum and 90 $\mu l$ of MeOH was added to reconstitute the sample to a final volume of about 100 $\mu l$.

### Chromatography and mass spectrometry

The chromatographic separation was performed on an Acquity UPLC BEH300 C4 column (100 × 1.0 mm; 1.7 $\mu m$ particle size; Waters Corporation) using a Waters Aquity UPLC system connected to an ABSCIEX 6500 QTRAP mass spectrometer for ion detection. All the samples were injected in duplicates and the flow rate was set to 100 $\mu l/min$. Solvent gradients were set, starting from 55% Solvent A (water + 0.1% formic acid)- 45% Solvent B (acetonitrile + 0.1% formic acid) from 0–5 min; then 45% B to 100% B from 5–10 min; 100% B was held from 10–15 min; between 15–16 min, 100% B was lowered to 45% B and held there till 20th min to re-equilibrate the column.

On the mass spectrometer, neutral loss scans were employed during pilot standardization experiments on biological samples to identify the parent ions that would lose neutral fragments corresponding to 490 a.m.u and 382 a.m.u—these were indicative of $PIP_2$ and PIP species respectively and likewise 155 a.m.u for PE species (Sharma et al, 2019). Thereafter, we quantified PIP, $PIP_2$, and PE species in biological samples using the selective MRM method in the positive ion mode. Area under the peaks was calculated via the Sciex MultiQuant software. For each run, area under the peak for each species of $PIP_2$, PIP, and PE was normalized to $PIP_2$, PIP and PE internal standard peak respectively. Thereafter, the sum of normalized areas for all the species of $PIP_2$ or PIP was taken and divided by the sum of normalized areas for all the species of PE in each of the biological samples to account for differences in total phospholipids extracted across samples. The MRM mass pairs used for PIP, $PIP_2$, and PE species identification and quantification are listed below in Table S4.

The other mass spectrometer parameters were as follows:

ESI voltage: +5,100–5200 V; source temperature: 300°C, curtain gas (CUR): 35–37, ion source gas 1 (GS1):15–20, ion source gas 2 (GS2): 15–20.

### RNA sequencing and transcriptomic analysis

RNA extracted from mature neurons (DIV45 with and without Li treatment) and mature neurons (DIV45 with and without CHIR99021 treatment) as mentioned above was submitted for paired-end sequencing. Quality check and sequencing was carried out at the NCBS Sanger sequencing facility using Illumina Hiseq 2500 system. Single-stranded RNA sequencing libraries were prepared for sequencing using NEBNext Ultra II Directional RNA Library Prep with Sample Purification Beads (Catalog no-E7765L). Paired-end reads, (125 base pair for each neuronal sample and 100 base pair for each NSC samples) was obtained after sequencing. Post adapter trimming, FastQC was used to check the quality of reads. The reads were then aligned to human genome (GRCh38) using hisat2-build and hisat2-align module. The module for single-strand sequencing was used in the Hisat-2 protocol. The output generated in the bam format, was further used to obtain the counts of each of the individual gene using the HT-Seq pipeline. The differential expression of genes was then calculated using DeSeq2. GO annotation analysis was done using the DAVID tool.

### Sampling and statistical analysis

Each experiment was performed unblinded on different biological groups with multiple biological replicates. No statistical analysis was done a priori to determine the sample sizes. For the calcium imaging experiments, quantification was from a minimum of 100 cells for all genotypes from multiple dishes of each type. For the $PIP_2$ probe experiments, ~50 cells were taken for the quantitative analysis. Statistical significance was computed using Student's unpaired $t$ test to check for differences in means between samples of different genotypes. Two-way ANOVA with post hoc Tukey's pairwise comparison was used for the grouped analysis. All statistical analyses were performed on Graph Pad Prism (version 9) and schematic representations were created with biorender.com.

## Supplementary Information

## Acknowledgements

This work was supported by the Department of Atomic Energy, Government of India, under Project Identification No. RTI 4006, the Department of Biotechnology, Government of India, through the Accelerator Program for Discovery in Brain Disorders (BT/PR17316/MED/31/326/2015), the Pratiksha Trust, and a Wellcome–DBT India Alliance Senior Fellowship to P Raghu (IA/S/14/2/501540) and Rohini Nilekani Philanthropies. We thank the NCBS Imaging, Mass spectrometry, Genomics, High performance computing, Stem cell, and Biosafety facilities for support.

### Author Contributions

S Saha: data curation, formal analysis, investigation, methodology, and writing—original draft, review, and editing.
H Krishnan: data curation, formal analysis, investigation, methodology, and writing—original draft, review, and editing.
P Raghu: conceptualization, supervision, funding acquisition, project administration, and writing—original draft, review, and editing.

### Conflict of Interest Statement

The authors declare that they have no conflict of interest.

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
