## [Reviewer comments · Life Science Alliance]

Life Science Alliance

IMPA1 dependent regulation of phosphatidylinositol 4,5 bisphosphate and calcium signaling by lithium

Padinjat Raghu, Sankhanil Saha, and Harini Krishnan

DOI: <https://doi.org/10.26508/lsa.202302425>

Corresponding author(s): Padinjat Raghu, National Centre for Biological Sciences

Review Timeline:	Submission Date:	2023-10-08
	Editorial Decision:	2023-10-10
	Revision Received:	2023-10-25
	Editorial Decision:	2023-11-16
	Revision Received:	2023-11-22
	Accepted:	2023-11-27

Transaction Report:

Please note that the manuscript was reviewed at *Review Commons* and these reports were taken into account in the decision-making process at *Life Science Alliance*.

Review
COMMONS

Review #1

Evidence, reproducibility and clarity

The authors functionally define in the inositol monophosphate phosphatase IMPA1, as the true target of lithium regulating phosphatidylinositol turnover and calcium signalling. While the observed IMPA1 inhibition by lithium led to the historical 'inositol depletion hypothesis' over the past 30+ years were published evidence both in support and against this concept. These contradictory sets of results have led to decreased interest in phosphoinositides as the signalling pathway affected by the therapeutic action of lithium in bipolar disorder (BD) patients. The remarkable results shown here will revert this trend since the data clearly demonstrate a key role of IMPA1 in setting the rate of phosphatidylinositol turnover, and consequentially the extent of calcium signalling. While the data are consistent with the 'inositol depletion hypothesis' the authors do not prove or disprove the validity of this hypothesis since the actual levels of inositol were not measured in their experiments. However, this is not a criticism, since quantifying cellular inositol is complex, it is just a suggestion for future work. After clarifying the points listed below this work will be suitable for publication.

The experiments using inositol rich DMEM (reported on page 17 and in Fig 2I,J) require a better explanation and an adequate material and method section. It is not clear if the 'normal/control' condition uses inositol-free DMEM. The standard concentration of inositol in DMEM is 40uM. Thus, are the ~155uM (28 mg/litre) added by the authors at the high end of the 40uM? Given that FBS contains inositol have the authors used dialyzed serum? While adding 155uM of inositol on either inositol-free medium or to medium containing 40uM inositol does not alter the author's message, this technical information are important for the reproducibility of the data presented and to understand how HEK293T manages inositol homeostasis.

It would be helpful to know if store operated calcium entry is altered in *impa1*^{-/-}M1 cells. This information would nicely complement Fig.3 C-E data.

In the Introduction at the end of page 4, the evidence not supporting the inositol depletion hypothesis is correctly discussed. This section lacks the discussion of another work questioning this theory (PMID: 30171184). The conclusion of this work is also in agreement with the authors finding that lithium affects the rare/turnover (lines 490/506) of PIP2 synthesis.

In the material and methods, Liquid Chromatography Mass spectrometry is abbreviated to LCMS while in the main text (line 493) LC-MS is used. The dashed version should be used throughout the manuscript.

I suggest to define (line 500) phosphatidylinositol 4-phosphate as (PI(4)P simplified as PIP). This will be consistent with the phosphatidylinositol 4,5-bisphosphate abbreviation as (PIP2) as reported in the introduction (line 97)

Line 646: Instead of using [this study] the authors should refer to the Figure panels supporting the discussed argument.

****Referees cross-commenting****

Reviewer #2 main message is identical to my. The work is a "contemporary re-evaluation of the inositol depletion hypothesis" but it does settle the debate. Say that reviewer #2 also recognises the importance of the work in defining IMPA1 as the only lithium target affecting the PI cycle removing GSK3 from the picture. Additionally, we agree that the thorough transcriptional analysis of the effect of lithium on human cortical neurons will be very informative for any researcher interested in psychiatric disorders.

Reviewer #2 requests are rational and not demanding. Most queries require extra information or the reformatting of the data presented.

2. Significance:

The submitted manuscript addresses an important topic. The authors developed HEK293 stable expressing muscarinic receptor to study the effect of lithium (without or after receptor activation) on PI(4,5)P2 turnover using two approaches, by microscopy and biochemically by LC-MS. These analyses were followed by a thorough characterization of the effect of lithium on calcium signalling. The generation of HEK293 impa1^{-/-} line has allowed the authors to demonstrate that the observed effect of lithium on PI(4,5)P2 turnover and calcium signalling were IMPA1 dependent. The authors pushed the work to a higher level by studying the effect of lithium on iPSC-derived human cortical neurons demonstrating that lithium reduces neurons excitability and calcium signalling. Although previously published attempts failed to generate IMPA1 deficient human cortical neurons the authors managed to produce iPSC impa1^{-/-} but, as reported and consistent with previous literature, this cell line failed to differentiate into neurons. This effort highlighted the author's commendable goal to develop a thoughtful story and not just another publication. The work is complemented by a very informative transcriptional analysis characterising the effect of lithium on human cortical neurons. Noteworthy is also the author's efforts to functionally and transcriptionally define the effect of another lithium target, GSK-3. These experiments emphasize that GSK-3 does not phenocopy the effect of lithium. This is another utterly important message of the paper.

In conclusion, the authors presented an easily readable, comprehensive, and experimentally convincing story. Furthermore, the developed experimental tools (HEK293-m1AChR) and the extensive data set (transcriptomic analysis) will be instrumental to further studies aimed at elucidating mechanistically how phosphoinositide signalling affects BD pathophysiology.

Review #2

1. Evidence, reproducibility and clarity:

This manuscript seeks to test if inhibition of the phosphoinositide (PI) cycle is the relevant pathway targeted by lithium in bipolar affective disorder (BPAD). Firstly, a cultured model system (HEK293T) is used to test the effects of lithium on the PI cycle. Using PI(4,5)P2 probes along with mass spectrometry, Li is shown to inhibit PI(4,5)P2 re-synthesis after PLC activation, though not to perturb pre-stimulus levels. Release of calcium from intracellular stores along with refilling from extracellular calcium is also inhibited - though there are no effects on stored calcium capacity. Crucially, with the exception of the calcium refilling step, these effects if Li can be abolished by genetic ablation of IMPA1, the proposed molecular target of Li. Having established the affected pathway, the manuscript then studies the effects of Li treatment on iPSC-differentiated cultured cortical neurons. Spontaneous and muscarinic evoked calcium transients are shown to be abolished by Li. None of these effects in HEK293T or neurons can be recapitulated by an inhibitor of GSK3beta, another proposed target for Li. Finally, a transcriptomic analysis of Li treated neurons is presented, showing down regulation of relevant genes, especially genes involved in neuronal calcium signaling and glutamatergic signaling.

The inositol depletion hypothesis has been debated for nearly four decades. As it stands, this manuscript does not settle this debate once and for all, but it does add some novel and important insights: that 1) IMPA1 is certainly the target of lithium, at least in terms of the PI cycle and 2) Lithium treatment can lead to longer-term transcriptional changes in neuronal calcium and glutamatergic signaling that can dampen excitability. The paper is on the whole clearly written, and the data are easy to follow. That said, there are a number of areas where the manuscript is lacking key details, or where the results do not fully support the conclusions. Specific suggestions for amendment are as follows:

1. The PH-PLCdelta1 PH domain has been used to follow PI(4,5)P2 turnover in HEK293T cells. Although long established, the manuscript does not discuss the fact that this domain also binds to IP3, which given high enough concentrations, can compete the PH domain off the membrane. As such, what is being measured is the convolution of PI(4,5)P2 decreases and IP3 increases (see for example doi: 10.1083/jcb.200301070). Ideally, a non-IP3 binding probe would have been used, such as the Tubby c-terminal domain (doi: 10.1186/1471-2121-10-67; doi: 10.1113/jphysiol.2008.153791). As it stands, the failure of the PH domain to return to the membrane after Li treatment reported in figures 1G, 3F and 3L

could either be due to a failure of PI(4,5)P2 re-synthesis, or a failure to breakdown IP3 - either of which are plausible explanations given inhibition of IMPA1. This concern is somewhat mitigated by the inclusion of mass determinations of the lipids in figure 1H-J, which support the PI(4,5)P2 re-synthesis defect. However, the potential problems with interpretation of the data with the PH domain should be discussed.

2. The strongest evidence for the effects of IMPA1 inhibition coming from inositol depletion are given by the experiment reported in figure 2I and J, where inositol supplementation rescues calcium mobilization. This should also be performed for the PIP2 re-synthesis experiments.

3. It is implicit in the manuscript that DMEM does not contain inositol. This is not true; Life technologies' formulation for DMEM contains 40 micromol/l myo-inositol, which is sufficient to support activity of both proton/myo-inositol and sodium/myo-inositol symporters (HMIT and SMIT). On the face of it, therefore, inositol depletion seems unlikely. The reviewer wonders what concentration of added inositol mediated the rescue? This key fact is missing from the manuscript. At the very least, the details should be included and the reason for rescue of already inositol replete cells discussed. Ideally, the key experiments would be repeated with inositol-free medium and supplementation.

4. The introduction refers to lithium as a "non-competitive" inhibitor of IMPA1. This is erroneous, as lithium is in fact an uncompetitive inhibitor. This is a key distinction: since the uncompetitive inhibitor blocks the enzyme:substrate complex, it is most effective where substrate accumulates the most - in this context, sites of intense PLC activity. This was central to Berridge's inositol depletion hypothesis. Also, the Allison et al citation is incorrect here. The correct citation is PMID: 2833231.

5. other key experimental details are missing from the figures/figure legends/results and or methods. Namely, what concentration of carbachol was used? What was the optimum concentration of thapsigargin? For figure 2 B-C, was carbachol used to evoke calcium mobilization?

6. The effects of IMPA1 knockout and rescue in figure 3F are rather unconvincing. All treatment groups' means fall within 1 SD; are the changes statistically significant? Plotting 95% C.I. or standard error may be more informative for these experiments.

****Minor comments:****

- There are some inconsistencies in the figure panels. Arrows labelled "CCh" are used to denote CCh addition in e.g. Fig. 3D, whereas simple arrows are used elsewhere e.g. Fig. 3F or no arrow at all e.g. Fig. 3B.

- Details of what the error denotes is missing in Figs. 2B-C, 3B-C, F - as is N for 3A.

- Fig. 2D: It should be made explicit that arrows indicate TTX addition in the figure. More importantly, it should be clarified whether this is also the case for Fig. 5D? Transients do not appear to be depleted by this addition.

- In figure 6C, it is stated that "there was no down regulation in transcripts for SCN1A (Nav1.1) or SCN9A (Nav7.1)" but this is not strictly true from the data; the Li-treated cells definitely trend lower. The effect is clearly not statistically significant, so although it is not possible to state that they reduced, this is not the same thing as being able to assert that they are not reduced. Perhaps it could be more helpful to plot the size of the reduction between these transcripts?

2. Significance:

The manuscript is ultimately a contemporary re-evaluation of the inositol depletion hypothesis for Li treatment of BPAD, first proposed by Berridge in 1989. The manuscript certainly does not end the debate - other pathways and mechanisms for lithium's actions on a complex human behavioral phenotype will surely persist. However, it does add some important new insights: firstly, that IMPA1 is certainly the target mediating the effects of Li on the PI cycle; and secondly, that long-term, Li may exert effects at the transcriptional level to down regulate calcium and glutamatergic signaling in the brain. However, there is no mechanism presented to link these two findings. The work is therefore of interest to researchers with an interest on studying psychiatric disorders, basic mechanisms of neuronal excitability as well as molecular mechanisms of cell signaling. It therefore deserves to be published in a journal with a cross disciplinary focus.

October 10, 2023

Re: Life Science Alliance manuscript #LSA-2023-02425-T

Prof. Raghu Padinjat
National Centre for Biological Sciences
Cellular Organization and Signalling
TIFR GKVK Campus
Bangalore, Karnataka 560065
India

Dear Dr. Padinjat,

Thank you for submitting your manuscript entitled "IMPA1 dependent regulation of plasma membrane phosphatidylinositol 4,5-bisphosphate turnover and calcium signalling by lithium" to Life Science Alliance. We invite you to re-submit the manuscript, revised according to your Revision Plan.

Thank you for this interesting contribution to Life Science Alliance. We are looking forward to receiving your revised manuscript.

Sincerely,

B. MANUSCRIPT ORGANIZATION AND FORMATTING:

1. General Statements

We thank the reviewer to appreciating the depth and quality of the results presented in our manuscript. We are happy that he/she has appreciated our contemporary approach to addressing an important problem but also a thorny and debated topic in this field that has lasted for over 30 years since it was first proposed by Mike Berridge. Our work not only addresses mechanisms in cultured human cells but also for the very first time addresses the key problem of Li action in the human brain using human iPSC derived forebrain cortical neurons.

We are delighted with the reviewer's comment that "This effort highlighted the author's commendable goal to develop a thoughtful story and not just another publication." We thank the reviewer for appreciating our detailed multi-disciplinary approach to addressing a long standing (> 4 decades) problem of key significance in human psychiatry and neuroscience. We thank him/her for noting that the work deserves to be presented in a journal with a cross-disciplinary focus. A few technical points that have been noted by the reviewer have been addressed below. Most of these can be effectively addressed by modest rewriting of text to make certain points clearer. The one experiment that has been suggested can be done and will be completed within 1 month.

2. Description of the planned revisions

Reviewer #1 (Evidence, reproducibility and clarity (Required)):

The authors functionally define in the inositol monophosphate phosphatase IMPA1, as the true target of lithium regulating phosphatidylinositol turnover and calcium signalling. While the observed IMPA1 inhibition by lithium led to the historical 'inositol depletion hypothesis' over the past 30+ years were published evidence both in support and against this concept. These contradictory sets of results have led to decreased interest in phosphoinositides as the signalling pathway affected by the therapeutic action of lithium in bipolar disorder (BD) patients. The remarkable results shown here will revert this trend since the data clearly demonstrate a key role of IMPA1 in setting the rate of phosphatidylinositol turnover, and consequentially the extent of calcium signalling. While the data are consistent with the 'inositol depletion hypothesis' the

authors do not prove or disprove the validity of this hypothesis since the actual levels of inositol were not measured in their experiments. However, this is not a criticism, since quantifying cellular inositol is complex, it is just a suggestion for future work. After clarifying the points listed below this work will be suitable for publication.

- The experiments using inositol rich DMEM (reported on page 17 and in Fig 2I,J) require a better explanation and an adequate material and method section. It is not clear if the 'normal/control' condition uses inositol-free DMEM. **The standard concentration of inositol in DMEM is 40uM. Thus, are the ~155uM (28 mg/litre) added by the authors at the high end of the 40uM?** Given that FBS contains inositol have the authors used dialyzed serum? While adding 155uM of inositol on either inositol-free medium or to medium containing 40uM inositol does not alter the author's message, this technical information are important for the reproducibility of the data presented and to understand how HEK293T manages inositol homeostasis.

The standard concentration of inositol in DMEM high Glucose media (Dulbecco's Modified Eagle Medium; Life Technologies) is 40 μ M (7 mg/ litre) and our HEK293T cells were maintained under standard conditions at (37°C, with 5% CO₂) in this media, supplemented with 10% Foetal bovine serum (FBS). This FBS was not dialyzed, so it might contain trace amounts of inositol.

For our inositol rich DMEM, 117 μ l of 100 μ M of inositol (18 mg/ml) was added to 100 ml of DMEM high Glucose media, supplemented with 10% FBS- this led to the final effective concentration of inositol in the media ~155 μ M (28 mg/ litre). This media was referred to as the inositol rich media and used for the inositol supplementation in Fig. 2I-J.

Therefore, the inositol supplementation we refer to is effectively raising the extracellular inositol concentration from ca. 40mM to 155mM which is 3X elevation.

This information has been added to the results section.

- It would be helpful to know if store operated calcium entry is altered in *impa1*^{-/-}M1 cells. This information would nicely complement Fig.3 C-E data.

We have studied store operated calcium entry in *IMPA1*^{-/-} cells and it is decreased with respect to the Control-M1 cells.

The quantification of this reduction can be added to the paper (Supp Fig 3E).

- In the Introduction at the end of page 4, the evidence not supporting the inositol depletion hypothesis is correctly discussed. This section lacks the discussion of another work questioning this theory (PMID: 30171184). The conclusion of this work is also in agreement with the authors finding that lithium affects the rare/turnover (lines 490/506) of PIP2 synthesis.

Saiardi A, Mudge AW. Lithium and fluoxetine regulate the rate of phosphoinositide synthesis in neurons: a new view of their mechanisms of action in bipolar disorder. *Transl Psychiatry*. 2018 Aug 31;8(1):175. doi: 10.1038/s41398-018-0235-2. PMID: 30171184.

This paper suggests that lithium mediated inhibition of IMPase leads to an accumulation of IP₁ and this elevated IP₁ leads to a competitive inhibition of the rate of synthesis of PI, and hence turnover of PIP₂. Combined with lithium's inhibition of inositol uptake, this inhibition of PI synthesis can lead to the mood stabilizing effect of lithium, rather than the inositol depletion.

This point has been added to our manuscript and the reference cited.

- In the material and methods, Liquid Chromatography Mass spectrometry is abbreviated to LCMS while in the main text (line 493) LC-MS is used. The dashed version should be used throughout the manuscript.

Liquid Chromatography Mass spectrometry has been abbreviated to LC-MS throughout the text in the revised version.

- I suggest to define (line 500) phosphatidylinositol 4-phosphate as (PI(4)P simplified as PIP). This will be consistent with the phosphatidylinositol 4,5-bisphosphate abbreviation as (PIP₂) as reported in the introduction (line 97)

PIP refers to all the functional isoforms of Phosphatidylinositol phosphate- PI 3P, PI 4P and PI 5P. By the LC-MS/MS analysis, we had measured the total PIP masses but we cannot distinguish between the individual functional isomers of PIP. However, pre-existing literature suggests that PI 4P is the most abundant isoform of PIP present in cell- its level is approximately 50 folds higher than that of PI 5P (*Rameh et al., 1997*). Hence we can suggest that the change in the total mass of PIP (as seen by the LC-MS/MS) is mainly reflective of the PI 4P.

This point has been added to the revised version of our manuscript.

- Line 646: Instead of using [this study] the authors should refer to the Figure panels supporting the discussed argument.

The identity of the channels mediating Ca²⁺ transients in this system was shown by us in *Sharma et al., 2020*.

In the revision, we have cited this paper, along with the information.

****Referees cross-commenting****

Reviewer #2 main message is identical to my. The work is a "contemporary re-evaluation of the inositol depletion hypothesis" but it does settle the debate. Say that reviewer #2 also recognises the importance of the work in defining IMPA1 as the only lithium target affecting the PI cycle removing GSK3 from the picture. Additionally, we agree that the thorough transcriptional analysis of the effect of lithium on human cortical neurons will be very informative for any researcher interested in psychiatric disorders.

Reviewer #2 requests are rational and not demanding. Most queries require extra information or the reformatting of the data presented.

Reviewer #1 (Significance (Required)):

The submitted manuscript addresses an important topic. The authors developed HEK293 stable expressing muscarinic receptor to study the effect of lithium (without or after receptor activation) on PI(4,5)P2 turnover using two approaches, by microscopy and biochemically by LC-MS. These analyses were followed by a thorough characterization of the effect of lithium on calcium signalling. The generation of HEK293 *impa1*^{-/-} line has allowed the authors to demonstrate that the observed effect of lithium on PI(4,5)P2 turnover and calcium signalling were IMPA1 dependent. The authors pushed the work to a higher level by studying the effect of lithium on iPSC-derived human cortical neurons demonstrating that lithium reduces neurons excitability and calcium signalling. Although previously published attempts failed to generate IMPA1 deficient human cortical neurons the authors managed to produce iPSC *impa1*^{-/-} but, as reported and consistent with previous literature, this cell line failed to differentiate into neurons. This effort highlighted the author's commendable goal to develop a thoughtful story and not just another publication. The work is complemented by a very informative transcriptional analysis characterising the effect of lithium on human cortical neurons. Noteworthy is also the author's efforts to functionally and transcriptionally define the effect of another lithium target, GSK-3. These experiments emphasize that GSK-3 does not phenocopy the effect of lithium. This is another utterly important message of the paper.

In conclusion, the authors presented an easily readable, comprehensive, and experimentally convincing story. Furthermore, the developed experimental tools (HEK293-m1AChR) and the extensive data set (transcriptomic analysis) will be instrumental to further studies aimed at elucidating mechanistically how phosphoinositide signalling affects BD pathophysiology.

Reviewer #2 (Evidence, reproducibility and clarity (Required)):

This manuscript seeks to test if inhibition of the phosphoinositide (PI) cycle is the relevant pathway targeted by lithium in bipolar affective disorder (BPAD). Firstly, a cultured model system (HEK293T) is used to test the effects of lithium on the PI cycle. Using PI(4,5)P2 probes along with mass spectrometry, Li is shown to inhibit PI(4,5)P2 re-synthesis after PLC activation, though not to perturb pre-stimulus levels. Release of calcium from intracellular stores along with refilling from extracellular calcium is also inhibited - though there are no effects on stored calcium capacity. Crucially, with the exception of the calcium refilling step, these effects if Li can be abolished by genetic ablation of IMPA1, the proposed molecular target of Li. Having established the affected pathway, the manuscript then studies the effects of Li treatment on iPSC-differentiated cultured cortical neurons. Spontaneous and muscarinic evoked calcium transients are shown to be abolished by Li. None of these effects in HEK293T or neurons can be recapitulated by an inhibitor of GSK3beta, another proposed target for Li. Finally, a transcriptomic analysis of Li treated neurons is presented, showing down regulation of relevant genes, especially genes involved in neuronal calcium signaling and glutamatergic signaling.

The inositol depletion hypothesis has been debated for nearly four decades. As it stands, this manuscript does not settle this debate once and for all, but it does add some novel and important insights: that 1) IMPA1 is certainly the target of lithium, at least in terms of the PI cycle and 2)

Lithium treatment can lead to longer-term transcriptional changes in neuronal calcium and glutamatergic signaling that can dampen excitability. The paper is on the whole clearly written, and the data are easy to follow. That said, there are a number of areas where the manuscript is lacking key details, or where the results do not fully support the conclusions. Specific suggestions for amendment are as follows:

(1) The PH-PLC δ 1 PH domain has been used to follow PI(4,5)P₂ turnover in HEK293T cells. Although long established, the manuscript does not discuss the fact that this domain also binds to IP₃, which given high enough concentrations, can compete the PH domain off the membrane. As such, what is being measured is the convolution of PI(4,5)P₂ decreases and IP₃ increases (see for example doi: 10.1083/jcb.200301070). Ideally, a non-IP₃ binding probe would have been used, such as the Tubby c-terminal domain (doi: 10.1186/1471-2121-10-67; doi: 10.1113/jphysiol.2008.153791). As it stands, the failure of the PH domain to return to the membrane after Li treatment reported in figures 1G, 3F and 3L could either be due to a failure of PI(4,5)P₂ re-synthesis, or a failure to breakdown IP₃ - either of which are plausible explanations given inhibition of IMPA1. This concern is somewhat mitigated by the inclusion of mass determinations of the lipids in figure 1H-J, which support the PI(4,5)P₂ re-synthesis defect. However, the potential problems with interpretation of the data with the PH domain should be discussed.

PH-PLC δ -GFP probe is used in the field as a biosensor for PI 4,5-P₂ (Chakrabarti et al., 2015; Várnai and Balla, 1998) and has been used to monitor the PIP₂ turn-over rate. However, as the reviewer has pointed out, this probe also has an affinity towards IP₃ (Xu et al., 2003) and therefore the failure of the probe to return to the plasma membrane could also, in principle, reflect the accumulation of IP₃.

We are well aware of this discussion in the field and to make sure that our measurements using the PH-PLC δ -GFP probe are indeed a true reflection of PIP₂ re-synthesis, we have also used a biochemical method to establish the levels of PIP₂. Our measurements of the total mass of PIP₂ by LC-MS/MS corroborate our findings using the probe, on the delay in PIP₂ resynthesis. Nonetheless, we will explicitly mention this point in the discussion.

Drawback of the Tubby c-terminal domain-

Most of the biosensors for PI 4,5-P₂ have distinctive advantage and disadvantages- PH-PLC δ -GFP probe is a more sensitive reporter but its IP₃ binding may compromise its accuracy to measure PI 4,5-P₂ changes. However, the Tubby c-terminal domain has exhibited lower sensitivity to report on changes of PI 4,5-P₂ during PLC activation, although being more specific in its affinity towards PI 4,5-P₂ (Szentpetery et al., 2009). Furthermore, recent studies have revealed that Tubby c-terminal domain can also bind to PI 3,4-P₂ as well as PI 3,4,5-P₃ (Hammond and Balla, 2015). Lastly a very recent study has noted that in contrast to PH-PLC δ -GFP probe, the tubby domain binds selectively to certain domains of the plasma membrane at membrane contact sites making it not a detector of PIP₂ levels across the plasma membrane (Thallmir et al., 2023, PMID: [PMC10445746](https://pubmed.ncbi.nlm.nih.gov/41445746/) DOI: [10.1242/jcs.260848](https://doi.org/10.1242/jcs.260848)).

(2) The strongest evidence for the effects of IMPA1 inhibition coming from inositol depletion are given by the experiment reported in figure 2I and J, where inositol supplementation rescues calcium mobilization. This should also be performed for the PIP₂ re-synthesis experiments.

We thank the reviewer for this suggestion. We have carried out this experiment- we have observed that rate of PLC induced PIP₂ regeneration that is reduced due to Li, is rescued by inositol supplementation in the media.

This result has been added to the revised manuscript (Supp Fig. 2D).

(3) It is implicit in the manuscript that DMEM does not contain inositol. This is not true; Life technologies' formulation for DMEM contains 40 micromol/l myo-inositol, which is sufficient to support activity of both proton/myo-inositol and sodium/myo-inositol symporters (HMIT and SMIT). On the face of it, therefore, inositol depletion seems unlikely. The reviewer wonders what concentration of added inositol mediated the rescue? This key fact is missing from the manuscript. At the very least, the details should be included and the reason for rescue of already inositol replete cells discussed. Ideally, the key experiments would be repeated with inositol-free medium and supplementation.

Repeated cycles of GPCR linked PLC signalling depend on a stable and on a continuous supply of PIP₂ at the cell membrane, which in turn depends on the cytoplasmic pool of inositol in the cell. Inositol pool can be maintained by three avenues- via the recycling of the inositol by the stepwise dephosphorylation of IP₃, *de novo* synthesis of inositol from glucose 6-phosphate and the transport of inositol from the extracellular environment across the plasma membrane. Li inhibits IMPase, an enzyme that dephosphorylates inositol monophosphate to generate free inositol. Due to Li's inhibition of IMPase, the inositol pool cannot be regenerated by the first two avenues since both of them need IMPase. However, restriction of the inositol pool by Li's inhibition of IMPase can be bypassed by the transport of inositol from the extracellular media via SMIT (Sodium-dependent myo-inositol co-transporter) and/or HMIT (Proton-dependent myo-inositol co-transporter). At steady state, the low amount of inositol in the DMEM media (*the inositol concentration in normal DMEM media being approximately 7 mg/litre or 40 μM*) might be sufficient for maintaining the inositol pool and thereby PIP₂ levels at the steady state. But this amount of inositol in the DMEM media appeared to be limiting to sustain the inositol pool and thereby PIP₂ levels under conditions of hyperactivated PIP₂ cycle. This is likely the reason why Li mediated inositol depletion (by inhibition of IMPA1) leads to a decreased rate of PIP₂ synthesis at the plasma membrane as well as decreased PLC mediated Ca²⁺ release, in the background of activated PLC.

However, when the cells were grown in an inositol rich DMEM media (*inositol concentration is ca. 28 mg/litre or approximately 155 μM; which is similar to the inositol concentration in the cerebrospinal fluid (Swahn, 1985; Shetty et al., 1996)*); transport of extracellular inositol by SMIT/HMIT could sustain a continuous level of PIP₂ levels even in a PLC activated background. This explains the rescue of the decreased PLC mediated Ca²⁺ release phenotype in the control-M1 cells grown in inositol supplemented DMEM despite the Li treatment.

This section has been added to the Discussion section in the revised version.

(4) The introduction refers to lithium as a "non-competitive" inhibitor of IMPA1. This is erroneous, as lithium is in fact an uncompetitive inhibitor. This is a key distinction: since the uncompetitive inhibitor blocks the enzyme:substrate complex, it is most effective where substrate accumulates the most - in this context, sites of intense PLC activity. This was central to Berridge's inositol depletion hypothesis. Also, the Allison et al citation is incorrect here. The correct citation is PMID: 2833231.

Li inhibits IMPA1 in an uncompetitive manner.

This has been corrected in the revised version of the manuscript, with the appropriate references. Ref cited by reviewer- Gee NS, Ragan CI, Watling KJ, Aspley S, Jackson RG, Reid GG, Gani D, Shute JK. The purification and properties of myo-inositol monophosphatase from bovine brain. Biochem J. 1988 Feb 1;249(3):883-9. doi: 10.1042/bj2490883. PMID: 2833231; PMCID: PMC1148789.

Ref- Hallcher LM, Sherman WR. The effects of lithium ion and other agents on the activity of myo-inositol-1-phosphatase from bovine brain. J Biol Chem. 1980 Nov 25;255(22):10896-901. PMID: 6253491.

(5) other key experimental details are missing from the figures/figure legends/results and or methods. Namely, what concentration of carbachol was used? What was the optimum concentration of thapsigargin? For figure 2 B-C, was carbachol used to evoke calcium mobilization?

For the agonist mediated Ca²⁺ release, 20 μM of carbachol was used. This is mentioned in the Materials and Methods section (line 238); this is mentioned in the figure legends and results for clarity, in the revised version.

For the store depletion in the SOCE experiments, 10 μM of thapsigargin was used. This is mentioned in the Materials and Methods section (line 249); this is mentioned in the figure legends and results for clarity, in the revised version.

In Fig. 2B, C, Carbachol was used to evoke calcium mobilization in the cells. This is mentioned in the Results section (line 510)- this is mentioned in the figure legends and results for clarity.

(6) The effects of IMPA1 knockout and rescue in figure 3F are rather unconvincing. All treatment groups' means fall within 1 SD; are the changes statistically significant? Plotting 95% C.I. or standard error may be more informative for these experiments.

This has been addressed in the revised version. This figure has been plotted with 95% C.I.

Minor comments:

() There are some inconsistencies in the figure panels. Arrows labelled "CCh" are used to denote CCh addition in e.g. Fig. 3D, whereas simple arrows are used elsewhere e.g. Fig. 3F or no arrow at all e.g. Fig. 3B.

All the points where CCh has been added to stimulate PLC have been denoted with an arrow labelled Cch in the figures for clarity and consistency.

This has been addressed in the figures of the revised version of the manuscript.

() Details of what the error denotes is missing in Figs. 2B-C, 3B-C, F - as is N for 3A.

Whiskers in box plots show the minimum and maximum values with a line at the median.

This has been added to the legends section addressed in the revised version.

() Fig. 2D: It should be made explicit that arrows indicate TTX addition in the figure. More importantly, it should be clarified whether this is also the case for Fig. 5D? Transients do not appear to be depleted by this addition.

This has been mentioned in the figure legends for the figure (line 1077)- we have addressed this in the revised version.

Few of the neuronal transients are not abolished by TTX- this variability has been addressed by other representative traces in the revised version.

() In figure 6C, it is stated that "there was no down regulation in transcripts for SCN1A (Nav1.1) or SCN9A (Nav7.1)" but this is not strictly true from the data; the Li-treated cells definitely trend lower. The effect is clearly not statistically significant, so although it is not possible to state that they reduced, this is not the same thing as being able to assert that they are not reduced. Perhaps it could be more helpful to plot the size of the reduction between these transcripts?

This has been addressed as- "there was no significant downregulation in transcripts for SCN1A (Nav1.1) or SCN9A (Nav7.1)" in the revised version of the manuscript.

The difference in the transcript level for SCN1A (Nav1.1) or SCN9A (Nav7.1) can be plotted for further clarification.

Reviewer #2 (Significance (Required)):

The manuscript is ultimately a contemporary re-evaluation of the inositol depletion hypothesis for Li treatment of BPAD, first proposed by Berridge in 1989. The manuscript certainly does not end the debate - other pathways and mechanisms for lithium's actions on a complex human behavioral phenotype will surely persist. However, it does add some important new insights: firstly, that IMPA1 is certainly the target mediating the effects of Li on the PI cycle; and secondly, that long-term, Li may exert effects at the transcriptional level to down regulate calcium and glutamatergic signaling in the brain. However, there is no mechanism presented to link these two findings. The work is therefore of interest to researchers with an interest on studying psychiatric disorders, basic

mechanisms of neuronal excitability as well as molecular mechanisms of cell signaling. It therefore deserves to be published in a journal with a cross disciplinary focus.

November 16, 2023

RE: Life Science Alliance Manuscript #LSA-2023-02425-TR

Prof. Padinjat Raghu
National Centre for Biological Sciences
Cellular Organization and Signalling
TIFR GKVK Campus
Bangalore, Karnataka 560065
India

Dear Dr. Raghu,

Thank you for submitting your revised manuscript entitled "IMPA1 dependent regulation of phosphatidylinositol 4,5 bisphosphate and calcium signaling by lithium". We would be happy to publish your paper in Life Science Alliance pending final revisions necessary to meet our formatting guidelines.

- please upload all figure files as individual ones, including the supplementary figure files; all figure legends should only appear in the main manuscript file
- please add the Twitter handle of your host institute/organization as well as your own or/and one of the authors in our system
- please consult our manuscript preparation guidelines <https://www.life-science-alliance.org/manuscript-prep> and make sure your manuscript sections are in the correct order
- please add an Author Contributions section to your main manuscript text
- please add your main, supplementary figure, and table legends to the main manuscript text after the references section
- figure S6 should be figure S5 -- please correct (figure label, legend, and call-outs)
- please add callouts for Figures S1B-D; S2A,B; S3G and S5C-E to your main manuscript text
- please label the table in the text as Table 1 and be sure to cite it in the text

A. FINAL FILES:

B. MANUSCRIPT ORGANIZATION AND FORMATTING:

Sincerely,

Reviewer #1 (Comments to the Authors (Required)):

The revised version of the manuscript submitted to Life Science Alliance (LSA) fully addresses my concerns expressed in my initial assessment for Review Commons. Specifically, the authors provided the requested technical details on the high inositol supplementation experimental condition. More importantly, they also performed experiments demonstrating that store-operated calcium entry is altered in *impa1^{-/-}M1* cells. All the other minor points raised have been clarified in the new version of the manuscript.

In my opinion, the authors also addressed reviewer #2 initial concerns.

Reviewer #2 (Comments to the Authors (Required)):

The authors have addressed my comments in their revision plan. I think the manuscript is an important advance and should be published in LSA.

November 27, 2023

RE: Life Science Alliance Manuscript #LSA-2023-02425-TRR

Prof. Padinjat Raghu
National Centre for Biological Sciences
Cellular Organization and Signalling
TIFR GKVK Campus
Bangalore, Karnataka 560065
India

Dear Dr. Raghu,

Thank you for submitting your Research Article entitled "IMPA1 dependent regulation of phosphatidylinositol 4,5 bisphosphate and calcium signaling by lithium". It is a pleasure to let you know that your manuscript is now accepted for publication in Life Science Alliance. Congratulations on this interesting work.

DISTRIBUTION OF MATERIALS:

Again, congratulations on a very nice paper. I hope you found the review process to be constructive and are pleased with how the manuscript was handled editorially. We look forward to future exciting submissions from your lab.

Sincerely,
